# PYRAFORMER: LOW-COMPLEXITY PYRAMIDAL ATTENTION FOR LONG-RANGE TIME SERIES MODELING AND FORECASTING

**Shizhan Liu**[1,2*]**, Hang Yu**[1*]**, Cong Liao**[1]**, Jianguo Li**[1†]**, Weiyao Lin**[2]**, Alex X. Liu**[1]**,
and Schahram Dustdar**[3]

[1]Ant Group, [2]Shanghai Jiaotong University, [3] TU Wien, Austria

## ABSTRACT

Accurate prediction of the future given the past based on time series data is of paramount importance, since it opens the door for decision making and risk management ahead of time. In practice, the challenge is to build a flexible but parsimonious model that can capture a wide range of temporal dependencies. In this paper, we propose Pyraformer by exploring the multi-resolution representation of the time series. Specifically, we introduce the pyramidal attention module (PAM) in which the inter-scale tree structure summarizes features at different resolutions and the intra-scale neighboring connections model the temporal dependencies of different ranges. Under mild conditions, the maximum length of the signal traversing path in Pyraformer is a constant (i.e., $\mathcal{O}(1)$) with regard to the sequence length $L$, while its time and space complexity scale linearly with $L$. Extensive experimental results show that Pyraformer typically achieves the highest prediction accuracy in both single-step and long-range multi-step forecasting tasks with the least amount of time and memory consumption, especially when the sequence is long[1].

## 1 INTRODUCTION

Time series forecasting is the cornerstone for downstream tasks such as decision making and risk management. As an example, reliable prediction of the online traffic for micro-services can yield early warnings of the potential risk in cloud systems. Furthermore, it also provides guidance for dynamic resource allocation, in order to minimize the cost without degrading the performance. In addition to online traffic, time series forecasting has also found vast applications in other fields, including disease propagation, energy management, and economics and finance.

The major challenge of time series forecasting lies in constructing a powerful but parsimonious model that can compactly capture temporal dependencies of different ranges. Time series often exhibit both short-term and long-term repeating patterns (Lai et al., 2018), and taking them into account is the key to accurate prediction. Of particular note is the more difficult task of handling long-range dependencies, which is characterized by the length of the longest signal traversing path (see Proposition 2 for the definition) between any two positions in the time series (Vaswani et al., 2017). The shorter the path, the better the dependencies are captured. Additionally, to allow the models to learn these long-term patterns, the historical input to the models should also be long. To this end, low time and space complexity is a priority.

Unfortunately, the present state-of-the-art methods fail to accomplish these two objectives simultaneously. On one end, RNN (Salinas et al., 2020) and CNN (Munir et al., 2018) achieve a low time complexity that is linear in terms of the time series length $L$, yet their maximum length of the signal traversing path is $\mathcal{O}(L)$, thus rendering them difficult to learn dependencies between distant positions. On the other extreme, Transformer dramatically shortens the maximum path to be $\mathcal{O}(1)$

---

*Equal contribution. This work was done when Shizhan Liu was a research intern at Ant Group.
†Corresponding author

[1]Code is available at: https://github.com/alipay/Pyraformer

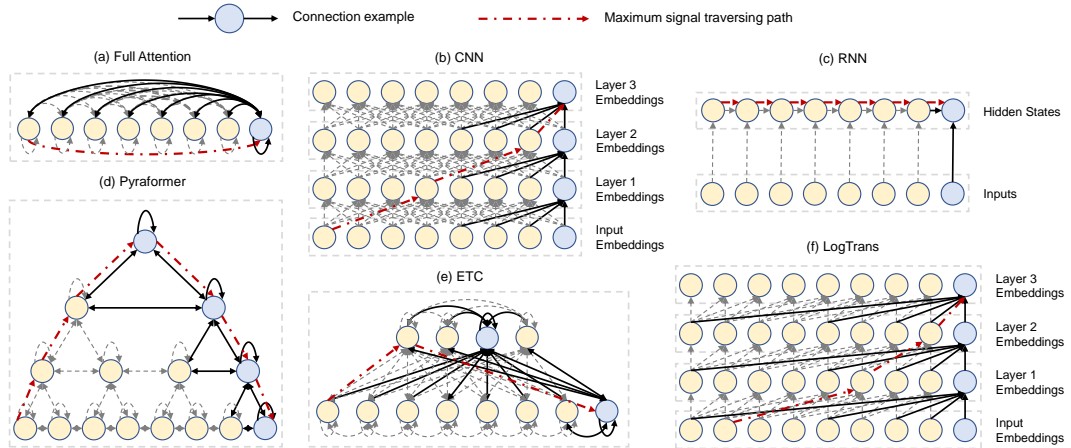

Figure 1: Graphs of commonly used neural network models for sequence data.

Table 1: Comparison of the complexity and the maximum signal traveling path for different models, where $G$ is the number of global tokens in ETC. In practice, the $G$ increases with $L$, and so the complexity of ETC is super-linear.

| Method | Complexity per layer | Maximum path length |
|---|---|---|
| CNN (Munir et al., 2018) | $\mathcal{O}(L)$ | $\mathcal{O}(L)$ |
| RNN (Salinas et al., 2020) | $\mathcal{O}(L)$ | $\mathcal{O}(L)$ |
| Full-Attention (Vaswani et al., 2017) | $\mathcal{O}(L^2)$ | $\mathcal{O}(1)$ |
| ETC (Ainslie et al., 2020) | $\mathcal{O}(GL)$ | $\mathcal{O}(1)$ |
| Longformer (Beltagy et al., 2020) | $\mathcal{O}(L)$ | $\mathcal{O}(L)$ |
| LogTrans (Li et al., 2019) | $\mathcal{O}(L \log L)$ | $\mathcal{O}(\log L)$ |
| Pyraformer | $\mathcal{O}(L)$ | $\mathcal{O}(1)$ |

at the sacrifice of increasing the time complexity to $\mathcal{O}(L^2)$. As a consequence, it cannot tackle very long sequences. To find a compromise between the model capacity and complexity, variants of Transformer are proposed, such as Longformer (Beltagy et al., 2020), Reformer (Kitaev et al., 2019), and Informer (Zhou et al., 2021). However, few of them can achieve a maximum path length less than $\mathcal{O}(L)$ while greatly reducing the time and space complexity.

In this paper, we propose a novel pyramidal attention based Transformer (Pyraformer) to bridge the gap between capturing the long-range dependencies and achieving a low time and space complexity. Specifically, we develop the pyramidal attention mechanism by passing messages based on attention in the pyramidal graph as shown in Figure 1(d). The edges in this graph can be divided into two groups: the inter-scale and the intra-scale connections. The inter-scale connections build a multiresolution representation of the original sequence: nodes at the finest scale correspond to the time points in the original time series (e.g., hourly observations), while nodes in the coarser scales represent features with lower resolutions (e.g., daily, weekly, and monthly patterns). Such latent coarser-scale nodes are initially introduced via a coarser-scale construction module. On the other hand, the intra-scale edges capture the temporal dependencies at each resolution by connecting neighboring nodes together. As a result, this model provides a compact representation for long-range temporal dependencies among far-apart positions by capturing such behavior at coarser resolutions, leading to a smaller length of the signal traversing path. Moreover, modeling temporal dependencies of different ranges at different scales with sparse neighboring intra-scale connections significantly reduces the computational cost. In short, our key contributions comprise:

- We propose Pyraformer to simultaneously capture temporal dependencies of different ranges in a compact multi-resolution fashion. To distinguish Pyraformer from the state-of-the-art methods, we summarize all models from the perspective of graphs in Figure 1.
- Theoretically, we prove that by choosing parameters appropriately, the maximum path length of $\mathcal{O}(1)$ and the time and space complexity of $\mathcal{O}(L)$ can be reached concurrently. To

highlight the appeal of the proposed model, we further compare different models in terms of the maximum path and the complexity in Table 1.

- Experimentally, we show that the proposed Pyraformer yields more accurate predictions than the original Transformer and its variants on various real-world datasets under the scenario of both single-step and long-range multi-step forecasting, but with lower time and memory cost.

## 2 RELATED WORKS

### 2.1 TIME SERIES FORECASTING

Time series forecasting methods can be roughly divided into statistical methods and neural network based methods. The first group involves ARIMA (Box & Jenkins, 1968) and Prophet (Taylor & Letham, 2018). However, both of them need to fit each time series separately, and their performance pales when it comes to long-range forecasting.

More recently, the development of deep learning has spawned a tremendous increase in neural network based time series forecasting methods, including CNN (Munir et al., 2018), RNN (Salinas et al., 2020) and Transformer (Li et al., 2019). As mentioned in the previous section, CNN and RNN enjoy a low time and space complexity (i.e., $\mathcal{O}(L)$), but entail a path of $\mathcal{O}(L)$ to describe long-range dependence. We refer the readers to Appendix A for a more detailed review on related RNN-based models. By contrast, Transformer (Vaswani et al., 2017) can effectively capture the long-range dependence with a path of $\mathcal{O}(1)$ steps, whereas the complexity increases vastly from $\mathcal{O}(L)$ to $\mathcal{O}(L^2)$. To alleviate this computational burden, LogTrans (Li et al., 2019) and Informer (Zhou et al., 2021) are proposed: the former constrains that each point in the sequence can only attend to the point that is $2^n$ steps before it, where $n = 1, 2, \cdots$, and the latter utilizes the sparsity of the attention score, resulting in substantial decrease in the complexity (i.e., $\mathcal{O}(L \log L)$ at the expense of introducing a longer maximum path length.

### 2.2 SPARSE TRANSFORMERS

In addition to the literature on time series forecasting, a plethora of methods have been proposed for enhancing the efficiency of Transformer in the field of natural language processing (NLP). Similar to CNN, Longformer (Beltagy et al., 2020) computes attention within a local sliding window or a dilated sliding window. Although the complexity is reduced to $\mathcal{O}(AL)$, where $A$ is the local window size, the limited window size makes it difficult to exchange information globally. The consequent maximum path length is $\mathcal{O}(L/A)$. As an alternative, Reformer (Kitaev et al., 2019) exploits locality sensitive hashing (LSH) to divide the sequence into several buckets, and then performs attention within each bucket. It also employs reversible Transformer to further reduce memory consumption, and so an extremely long sequence can be processed. Its maximum path length is proportional to the number of buckets though, and worse still, a large bucket number is required to reduce the complexity. On the other hand, ETC (Ainslie et al., 2020) introduces an extra set of global tokens for the sake of global information exchange, leading to an $\mathcal{O}(GL)$ time and space complexity and an $\mathcal{O}(1)$ maximum path length, where $G$ is the number of global tokens. However, $G$ typically increases with $L$, and the consequent complexity is still super-linear. Akin to ETC, the proposed Pyraformer also introduces global tokens, but in a multiscale manner, successfully reducing the complexity to $\mathcal{O}(L)$ without increasing the order of the maximum path length as in the original Transformer.

### 2.3 HIERARCHICAL TRANSFORMERS

Finally, we provide a brief review on methods that improve Transformer's ability to capture the hierarchical structure of natural language, although they have never been used for time series forecasting. HIBERT (Miculicich et al., 2018) first uses a Sent Encoder to extract the features of a sentence, and then forms the EOS tokens of sentences in the document as a new sequence and input it into the Doc Encoder. However, it is specialized for natural language and cannot be generalized to other sequence data. Multi-scale Transformer (Subramanian et al., 2020) learns the multi-scale representations of sequence data using both the top-down and bottom-up network structures. Such multi-scale representations help reduce the time and memory cost of the original Transformer, but

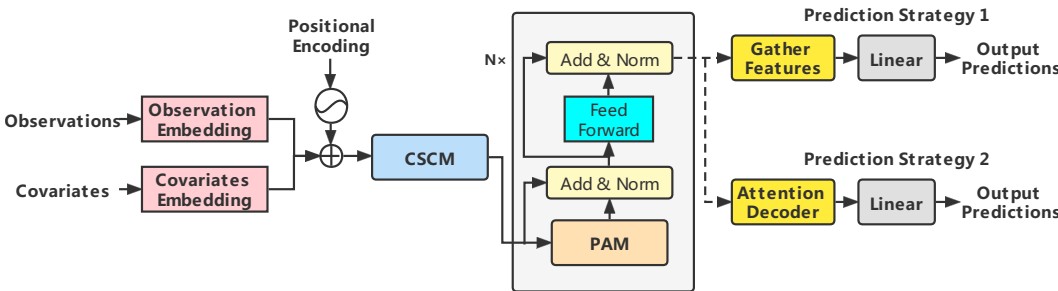

Figure 2: The architecture of Pyraformer: The CSCM summarizes the embedded sequence at different scales and builds a multi-resolution tree structure. Then the PAM is used to exchange information between nodes efficiently.

it still suffers from the pitfall of the quadratic complexity. Alternatively, BP-Transformer (Ye et al., 2019) recursively partitions the entire input sequence into two until a partition only contains a single token. The partitioned sequences then form a binary tree. In the attention layer, each upper-scale node can attend to its own children, while the nodes at the bottom scale can attend to the adjacent $A$ nodes at the same scale and all coarser-scale nodes. Note that BP-Transformer initializes the nodes at coarser scale with zeros, whereas Pyraformer introduces the coarser-scale nodes using a construction module in a more flexible manner. Moreover, BP-Transformer is associated with a denser graph than Pyraformer, thus giving rise to a higher complexity of $\mathcal{O}(L \log L)$.

## 3 METHOD

The time series forecasting problem can be formulated as predicting the future $M$ steps $\boldsymbol{z}_{t+1:t+M}$ given the previous $L$ steps of observations $\boldsymbol{z}_{t-L+1:t}$ and the associated covariates $\boldsymbol{x}_{t-L+1:t+M}$ (e.g., hour-of-the-day). To move forward to this goal, we propose Pyraformer in this paper, whose overall architecture is summarized in Figure 2. As shown in the figure, we first embed the observed data, the covariates, and the positions separately and then add them together, in the same vein with Informer (Zhou et al., 2021). Next, we construct a multi-resolution $C$-ary tree using the coarser-scale construction module (CSCM), where nodes at a coarser scale summarize the information of $C$ nodes at the corresponding finer scale. To further capture the temporal dependencies of different ranges, we introduce the pyramidal attention module (PAM) by passing messages using the attention mechanism in the pyramidal graph. Finally, depending on the downstream task, we employ different network structures to output the final predictions. In the sequel, we elaborate on each part of the proposed model. For ease of exposition, all notations in this paper are summarized in Table 4.

### 3.1 PYRAMIDAL ATTENTION MODULE (PAM)

We begin with the introduction of the PAM, since it lies at the heart of Pyraformer. As demonstrated in Figure 1(d), we leverage a pyramidal graph to describe the temporal dependencies of the observed time series in a multiresolution fashion. Such a multiresolution structure has proved itself an effective and efficient tool for long-range interaction modeling in the field of computer vision (Sun et al., 2019; Wang et al., 2021) and statistical signal processing (Choi et al., 2008; Yu et al., 2019). We can decompose the pyramidal graph into two parts: the inter-scale and the intra-scale connections. The inter-scale connections form a $C$-ary tree, in which each parent has $C$ children. For example, if we associate the finest scale of the pyramidal graph with hourly observations of the original time series, the nodes at coarser scales can be regarded as the daily, weekly, and even monthly features of the time series. As a consequence, the pyramidal graph offers a multi-resolution representation of the original time series. Furthermore, it is easier to capture long-range dependencies (e.g., monthly dependence) in the coarser scales by simply connecting the neighboring nodes via the intra-scale connections. In other words, the coarser scales are instrumental in describing long-range correlations in a manner that is graphically far more parsimonious than could be solely captured with a single, finest scale model. Indeed, the original single-scale Transformer (see Figure 1(a)) adopts a full graph that connects every two nodes at the finest scale so as to model the long-range dependencies, leading to a computationally burdensome model with $\mathcal{O}(L^2)$ time and space complexity (Vaswani

et al., 2017). In stark contrast, as illustrated below, the pyramidal graph in the proposed Pyraformer reduces the computational cost to $\mathcal{O}(L)$ without increasing the order of the maximum length of the signal traversing path.

Before delving into the PAM, we first introduce the original attention mechanism. Let $\boldsymbol{X}$ and $\boldsymbol{Y}$ denote the input and output of a single attention head respectively. Note that multiple heads can be introduced to describe the temporal pattern from different perspectives. $\boldsymbol{X}$ is first linearly transformed into three distinct matrices, namely, the query $\boldsymbol{Q} = \boldsymbol{X}\boldsymbol{W}_Q$, the key $\boldsymbol{K} = \boldsymbol{X}\boldsymbol{W}_K$, and the value $\boldsymbol{V} = \boldsymbol{X}\boldsymbol{W}_V$, where $\boldsymbol{W}_Q, \boldsymbol{W}_K, \boldsymbol{W}_V \in \mathbb{R}^{L \times D_K}$. For the $i$-th row $\boldsymbol{q}_i$ in $\boldsymbol{Q}$, it can attend to any rows (i.e., keys) in $\boldsymbol{K}$. In other words, the corresponding output $\boldsymbol{y}_i$ can be expressed as:

$$\boldsymbol{y}_i = \sum_{\ell=1}^{L} \frac{\exp(\boldsymbol{q}_i \boldsymbol{k}_\ell^T / \sqrt{D_K}) \boldsymbol{v}_\ell}{\sum_{\ell=1}^{L} \exp(\boldsymbol{q}_i \boldsymbol{k}_\ell^T / \sqrt{D_K})}, \tag{1}$$

where $\boldsymbol{k}_\ell^T$ denotes the transpose of row $\ell$ in $\boldsymbol{K}$. We emphasize that the number of query-key dot products (Q-K pairs) that need to be calculated and stored dictates the time and space complexity of the attention mechanism. Viewed another way, this number is proportional to the number of edges in the graph (see Figure 1(a)). Since all Q-K pairs are computed and stored in the full attention mechanism (1), the resulting time and space complexity is $\mathcal{O}(L^2)$.

As opposed to the above full attention mechanism, every node only pays attention to a limited set of keys in the PAM, corresponding to the pyramidal graph in Figure 1d. Concretely, suppose that $n_\ell^{(s)}$ denotes the $\ell$-th node at scale $s$, where $s = 1, \cdots, S$ represents the bottom scale to the top scale sequentially. In general, each node in the graph can attend to a set of neighboring nodes $\mathbb{N}_\ell^{(s)}$ at three scales: the adjacent $A$ nodes at the same scale including the node itself (denoted as $\mathbb{A}_\ell^{(s)}$), the $C$ children it has in the $C$-ary tree (denoted as $\mathbb{C}_\ell^{(s)}$), and the parent of it in the $C$-ary tree (denoted $\mathbb{P}_\ell^{(s)}$), that is,

$$\begin{cases} \mathbb{N}_\ell^{(s)} &= \mathbb{A}_\ell^{(s)} \cup \mathbb{C}_\ell^{(s)} \cup \mathbb{P}_l^{(s)} \\ \mathbb{A}_\ell^{(s)} &= \{n_j^{(s)} : |j - \ell| \leq \frac{A-1}{2}, 1 \leq j \leq \frac{L}{C^{s-1}}\} \\ \mathbb{C}_\ell^{(s)} &= \{n_j^{(s-1)} : (\ell-1)C < j \leq \ell C\} \quad \text{if } s \geq 2 \text{ else } \emptyset \\ \mathbb{P}_\ell^{(s)} &= \{n_j^{(s+1)} : j = \lceil \frac{\ell}{C} \rceil\} \quad \text{if } s \leq S-1 \text{ else } \emptyset \end{cases} \tag{2}$$

It follows that the attention at node $n_\ell^{(s)}$ can be simplified as:

$$\boldsymbol{y}_i = \sum_{\ell \in \mathbb{N}_\ell^{(s)}} \frac{\exp(\boldsymbol{q}_i \boldsymbol{k}_\ell^T / \sqrt{d_K}) \boldsymbol{v}_\ell}{\sum_{\ell \in \mathbb{N}_l^{(s)}} \exp(\boldsymbol{q}_i \boldsymbol{k}_\ell^T / \sqrt{d_K})}, \tag{3}$$

We further denote the number of attention layers as $N$. Without loss of generality, we assume that $L$ is divisible by $C^{S-1}$. We can then have the following lemma (cf. Appendix B for the proof and Table 4 for the meanings of the notations).

**Lemma 1.** *Given $A$, $C$, $L$, $N$, and $S$ that satisfy Equation* (4)*, after $N$ stacked attention layers, nodes at the coarsest scale can obtain a global receptive field.*

$$\frac{L}{C^{S-1}} - 1 \leq \frac{(A-1)N}{2}. \tag{4}$$

In addition, when the number of scales $S$ is fixed, the following two propositions summarize the time and space complexity and the order of the maximum path length for the proposed pyramidal attention mechanism. We refer the readers to Appendix C and D for proof.

**Proposition 1.** *The time and space complexity for the pyramidal attention mechanism is $\mathcal{O}(AL)$ for given $A$ and $L$ and amounts to $\mathcal{O}(L)$ when $A$ is a constant w.r.t. $L$.*

**Proposition 2.** *Let the signal traversing path between two nodes in a graph denote the shortest path connecting them. Then the maximum length of signal traversing path between two arbitrary nodes in the pyramidal graph is $\mathcal{O}(S + L/C^{S-1}/A)$ for given $A$, $C$, $L$, and $S$. Suppose that $A$ and $S$ are fixed and $C$ satisfies Equation* (5)*, the maximum path length is $\mathcal{O}(1)$ for time series with length $L$.*

$$\sqrt[s-1]{L} \geq C \geq \sqrt[s-1]{\frac{L}{(A-1)N/2 + 1}}. \tag{5}$$

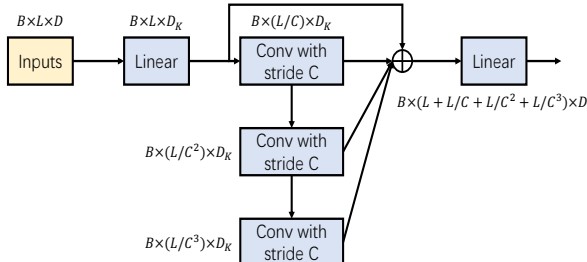

Figure 3: Coarser-scale construction module: $B$ is the batch size and $D$ is the dimension of a node.

In our experiments, we fix $S$ and $N$, and $A$ can only take 3 or 5, regardless of the sequence length $L$. Therefore, the proposed PAM achieves the complexity of $\mathcal{O}(L)$ with the maximum path length of $\mathcal{O}(1)$. Note that in the PAM, a node can attend to at most $A + C + 1$ nodes. Unfortunately, such a sparse attention mechanism is not supported in the existing deep learning libraries, such as Pytorch and TensorFlow. A naive implementation of the PAM that can fully exploit the tensor operation framework is to first compute the product between all Q-K pairs, i.e., $\boldsymbol{q}_i \boldsymbol{k}_\ell^T$ for $\ell = 1, \cdots, L$, and then mask out $\ell \notin \mathbb{N}_\ell^{(s)}$. However, the resulting time and space complexity of this implementation is still $\mathcal{O}(L^2)$. Instead, we build a customized CUDA kernel specialized for the PAM using TVM (Chen et al., 2018), practically reducing the computational time and memory cost and making the proposed model amenable to long time series. Longer historical input is typically helpful for improving the prediction accuracy, as more information is provided, especially when long-range dependencies are considered.

## 3.2 COARSER-SCALE CONSTRUCTION MODULE (CSCM)

CSCM targets at initializing the nodes at the coarser scales of the pyramidal graph, so as to facilitate the subsequent PAM to exchange information between these nodes. Specifically, the coarse-scale nodes are introduced scale by scale from bottom to top by performing convolutions on the corresponding children nodes $\mathbb{C}_\ell^{(s)}$. As demonstrated in Figure 3, several convolution layers with kernel size $C$ and stride $C$ are sequentially applied to the embedded sequence in the dimension of time, yielding a sequence with length $L/C^s$ at scale $s$. The resulting sequences at different scales form a $C$-ary tree. We concatenate these fine-to-coarse sequences before inputting them to the PAM. In order to reduce the amount of parameters and calculations, we reduce the dimension of each node by a fully connected layer before inputting the sequence into the stacked convolution layers and restore it after all convolutions. Such a bottleneck structure significantly reduces the number of parameters in the module and can guard against over-fitting.

## 3.3 PREDICTION MODULE

For single-step forecasting, we add an end token (by setting $z_{t+1} = 0$) to the end of the historical sequence $z_{t-L+1:t}$ before inputting it into the embedding layer. After the sequence is encoded by the PAM, we gather the features given by the last nodes at all scales in the pyramidal graph, concatenate and then input them into a fully connected layer for prediction.

For multi-step forecasting, we propose two prediction modules. The first one is the same with the single-step forecasting module, but maps the last nodes at all scales to all $M$ future time steps in a batch. The second one, on the other hand, resorts to a decoder with two full attention layers. Specifically, similar to the original Transformer (Vaswani et al., 2017), we replace the observations at the future $M$ time steps with 0, embed them in the same manner with the historical observations, and refer to the summation of the observation, covariate, and positional embedding as the "prediction token" $\boldsymbol{F}_p$. The first attention layer then takes the prediction tokens $\boldsymbol{F}_p$ as the query and the output of the encoder $\boldsymbol{F}_e$ (i.e., all nodes in the PAM) as the key and the value, and yields $\boldsymbol{F}_{d1}$. The second layer takes $\boldsymbol{F}_{d1}$ as the query, but takes the concatenated $\boldsymbol{F}_{d1}$ and $\boldsymbol{F}_e$ as the key and the value. The historical information $\boldsymbol{F}_e$ is fed directly into both attention layers, since such information is vital for accurate long-range forecasting. The final prediction is then obtained through a fully connected layer across the dimension of channels. Again, we output all future predictions together to avoid the problem of error accumulation in the autoregressive decoder of Transformer.

Table 2: Single-step forecasting results on three datasets. "Q-K pairs" refer to the number of query-key dot products performed by all attention layers in the network, which encodes the time and space complexity. We write the number of attention layers by $N$, the number of attention heads by $H$, the number of scales by $S$, the dimension of a node by $D$, the dimension of a key by $D_K$, the maximum dimension of feed-forward layer by $D_F$, and the convolution stride by $C$.

| Methods | Parameters | Datasets | NRMSE | ND | Q-K pairs |
|---|---|---|---|---|---|
| Full-attention | $\mathcal{O}(N(HDD_K+DD_F))$ | Electricity | 0.328 | 0.041 | 456976 |
| | | Wind | 0.175 | 0.082 | 589824 |
| | | App Flow | 0.407 | 0.080 | 589824 |
| LogTrans | $\mathcal{O}(N(HDD_K+DD_F))$ | Electricity | 0.333 | 0.041 | 50138 |
| | | Wind | 0.173 | 0.081 | 58272 |
| | | App Flow | 0.387 | 0.073 | 58272 |
| Reformer | $\mathcal{O}(N(HDD_K+DD_F))$ | Electricity | 0.359 | 0.047 | 677376 |
| | | Wind | 0.183 | 0.086 | 884736 |
| | | App Flow | 0.463 | 0.095 | 884736 |
| ETC | $\mathcal{O}(N(HDD_K+DD_F))$ | Electricity | 0.324 | 0.041 | 79536 |
| | | Wind | 0.167 | 0.074 | 102144 |
| | | App Flow | 0.397 | 0.069 | 102144 |
| Longformer | $\mathcal{O}(N(HDD_K+DD_F))$ | Electricity | 0.330 | 0.041 | 41360 |
| | | Wind | 0.166 | 0.075 | 52608 |
| | | App Flow | 0.377 | 0.07 | 52608 |
| Pyraformer | $\mathcal{O}(N(HDD_K + DD_F) +(S-1)CD_K^2)$ | Electricity | **0.324** | **0.041** | **17648** |
| | | Wind | **0.161** | **0.072** | **20176** |
| | | App Flow | **0.366** | **0.067** | **20176** |

## 4 EXPERIMENTS

### 4.1 DATASETS AND EXPERIMENT SETUP

We demonstrated the advantages of the proposed Pyraformer on the four real-world datasets, including Wind, App Flow, Electricity, and ETT. The first three datasets were used for single-step forecasting, while the last two for long-range multi-step forecasting. We refer the readers to Appendix E and F for more details regarding the data description and the experiment setup.

### 4.2 RESULTS AND ANALYSIS

#### 4.2.1 SINGLE-STEP FORECASTING

We conducted single-step prediction experiments on three datasets: Electricity, Wind and App Flow. The historical length is 169, 192 and 192, respectively, including the end token. We benchmarked Pyraformer against 5 other attention mechanisms, including the original full-attention (Vaswani et al., 2017), the log-sparse attention (i.e., LogTrans) (Li et al., 2019), the LSH attention (i.e., Reformer) (Kitaev et al., 2019), the sliding window attention with global nodes (i.e., ETC) (Ainslie et al., 2020), and the dilated sliding window attention (i.e., Longformer) (Beltagy et al., 2020). In particular for ETC, some nodes with equal intervals at the finest scale were selected as the global nodes. A global node can attend to all nodes across the sequence and all nodes can attend to it in turn(see Figure 1(e)). The training and testing schemes were the same for all models. We further investigated the usefulness of the pretraining strategy (see Appendix G), the weighted sampler, and the hard sample mining on all methods, and the best results were presented. We adopted the NRMSE (Normalized RMSE) and the ND (Normalized Deviation) as the evaluation indicators (see Appendix H for the definitions). The results are summarized in Table 2. For a fair comparison, except for full-attention, the overall dot product number of all attention mechanisms was controlled to the same order of magnitude.

Our experimental results show that Pyraformer outperforms Transformer and its variants in terms of NRMSE and ND, with the least number of query-key dot products (a.k.a. Q-K pairs). Con-

Table 3: Long-range multi-step forecasting results.

| Methods | Metrics | ETTh1 | | | ETTm1 | | | Electricity | | |
|---|---|---|---|---|---|---|---|---|---|---|
| | | 168 | 336 | 720 | 96 | 288 | 672 | 168 | 336 | 720 |
| Informer | MSE | 1.075 | 1.329 | 1.384 | 0.556 | 0.841 | 0.921 | 0.745 | 1.579 | 4.365 |
| | MAE | 0.801 | 0.911 | 0.950 | 0.537 | 0.705 | 0.753 | 0.266 | 0.323 | 0.371 |
| | Q-K pairs | 188040 | 188040 | 423360 | 276480 | 560640 | 560640 | 188040 | 188040 | 423360 |
| LogTrans | MSE | 0.983 | 1.100 | 1.411 | 0.554 | 0.786 | 1.169 | 0.791 | 1.584 | 4.362 |
| | MAE | 0.766 | 0.839 | 0.991 | 0.499 | 0.676 | 0.868 | 0.340 | 0.336 | 0.366 |
| | Q-K pairs | 74664 | 74664 | 216744 | 254760 | 648768 | 648768 | 74664 | 74664 | 216744 |
| Longformer | MSE | 0.860 | 0.975 | 1.091 | 0.526 | 0.767 | 1.021 | 0.766 | 1.591 | 4.361 |
| | MAE | 0.710 | 0.769 | 0.832 | 0.507 | 0.663 | 0.788 | 0.311 | 0.343 | 0.368 |
| | Q-K pairs | 63648 | 63648 | 249120 | 329760 | 1007136 | 1007136 | 63648 | 63648 | 249120 |
| Reformer | MSE | 0.958 | 1.044 | 1.458 | 0.543 | 0.924 | 0.981 | 0.783 | 1.584 | 4.374 |
| | MAE | 0.741 | 0.787 | 0.987 | 0.528 | 0.722 | 0.778 | 0.332 | 0.334 | 0.374 |
| | Q-K pairs | 1016064 | 1016064 | 2709504 | 5308416 | 14450688 | 14450688 | 1016064 | 1016064 | 2709504 |
| ETC | MSE | 1.025 | 1.084 | 1.137 | 0.762 | 1.227 | 1.272 | 0.777 | 1.586 | 4.361 |
| | MAE | 0.771 | 0.811 | 0.866 | 0.653 | 0.880 | 0.908 | 0.326 | 0.340 | 0.368 |
| | Q-K pairs | 125280 | 125280 | 288720 | 331344 | 836952 | 836952 | 125280 | 125280 | 288720 |
| Pyraformer | MSE | **0.808** | **0.945** | **1.022** | **0.480** | **0.754** | **0.857** | **0.719** | **1.533** | **4.312** |
| | MAE | **0.683** | **0.766** | **0.806** | **0.486** | **0.659** | **0.707** | **0.256** | **0.291** | **0.346** |
| | Q-K pairs | **26472** | **26472** | **74280** | **57264** | **96384** | **96384** | **26472** | **26472** | **74280** |

cretely, there are three major trends that can be gleaned from Table 2: (1) The proposed Pyraformer yields the most accurate prediction results, suggesting that the pyramidal graph can better explain the temporal interactions in the time series by considering dependencies of different ranges. Interestingly, for the Wind dataset, sparse attention mechanisms, namely, LogTrans, ETC, Longformer and Pyraformer, outperform the original full attention Transformer, probably because the data contains a large number of zeros and the promotion of adequate sparsity can help avoid over-fitting. (2) The number of Q-K pairs in Pyraformer is the smallest. Recall that this number characterizes the time and space complexity. Remarkably enough, it is $65.4\%$ fewer than that of LogTrans and $96.6\%$ than that of the full attention. It is worth emphasizing that this computational gain will continue to increase for longer time series. (3) The number of parameters for Pyraformer is slightly larger than that of the other models, resulting from the CSCM. However, this module is very lightweight, which incurs merely $5\%$ overhead in terms of model size compared to other models. Moreover, in practice, we can fix the hyper-parameters $A$, $S$ and $N$, and ensure that $C$ satisfies $C > \sqrt[S-1]{L/((A-1)N/2+1)}$. Consequently, the extra number of parameters introduced by the CSCM is only $\mathcal{O}((S-1)CD_K^2) \approx \mathcal{O}(\sqrt[S-1]{L})$.

### 4.2.2 LONG-RANGE MULTI-STEP FORECASTING

We evaluated the performance of Pyraformer for long-range forecasting on three datasets, that is, Electricity, ETTh1, and ETTm1. In particular for ETTh1 and ETTm1, we predicted the future oil temperature and the 6 power load features at the same time, which is a multivariate time series forecasting problem. Both prediction modules introduced in Section 3.3 were tested for all models and the better results are listed in Table 3.

It is evident that Pyraformer still achieves the best performance with the least number of Q-K pairs for all datasets regardless of the prediction length. More precisely, in comparison with Informer (Zhou et al., 2021), the MSE given by Pyraformer for ETTh1 is decreased by $24.8\%$, $28.9\%$, $26.2\%$ respectively when the prediction length is 168, 336, and 720. Once again, this bolsters our belief that it is more beneficial to employ the pyramidal graph when describing the temporal dependencies. Interestingly, we notice that for Pyraformer, the results given by the first prediction module are better than those by the second one. One possible explanation is that the second prediction module based on the full attention layers cannot differentiate features with different resolutions, while the first module based on a single fully connected layer can take full advantages of such features in an automated fashion. To better elucidate the modeling capacity of Pyraformer for long-range forecasting, we refer the readers to Appendix I for a detailed example on synthetic data.

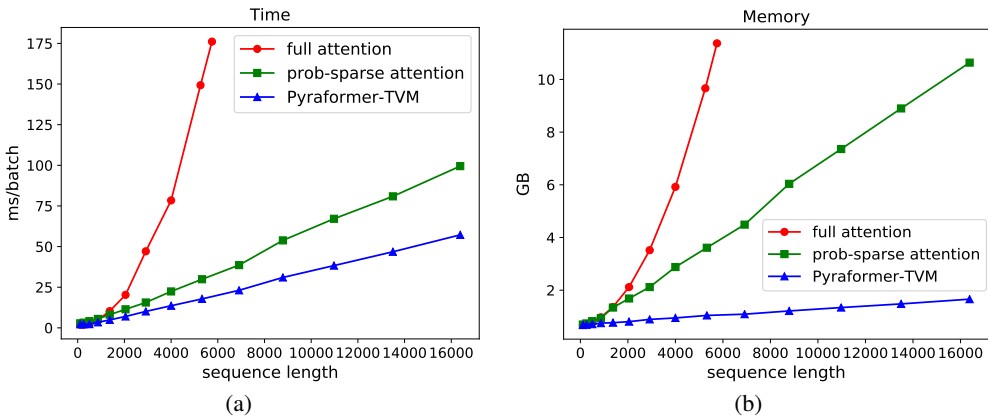

Figure 4: Comparison of the time and memory consumption between the full, the prob-sparse, and the TVM implementation of the pyramidal attention: (a) computation time; (b) memory occupation.

### 4.2.3 SPEED AND MEMORY CONSUMPTION

To check the efficiency of the customized CUDA kernel implemented based on TVM, we depicted the empirical computation time and memory cost as a function of the sequence length $L$ in Figure 4. Here we only compared Pyraformer with the full attention and the prob-sparse attention in Informer (Zhou et al., 2021). All the computations were performed on a 12 GB Titan Xp GPU with Ubuntu 16.04, CUDA 11.0, and TVM 0.8.0. Figure 4 shows that the time and memory cost of the proposed Pyraformer based on TVM is approximately a linear function of $L$, as expected. Furthermore, the time and memory consumption of the TVM implementation can be several orders of magnitude smaller than that of the full attention and the prob-sparse attention, especially for relatively long time series. Indeed, for a 12GB Titan Xp GPU, when the sequence length reaches 5800, full attention encounters the out-of-memory (OOM) problem, yet the TVM implementation of Pyraformer only occupies 1GB of memory. When it comes to a sequence with 20000 time points, even Informer incurs the OOM problem, whereas the memory cost of Pyraformer is only 1.91GB and the computation time per batch is only 0.082s.

### 4.3 ABLATION STUDY

We also performed ablation studies to measure the impact of $A$ and $C$, the CSCM architecture, the history length, and the PAM on the prediction accuracy of Pyraformer. The results are displayed in Tables 7-10. Detailed Discussions on the results can be found in Appendix J. Here, we only provide an overview of the major findings: (1) it is better to increase $C$ with $L$ but fix $A$ to a small constant for the sake of reducing the prediction error; (2) convolution with bottleneck strikes a balance between the prediction accuracy and the number of parameters, and hence, we use it as the CSCM; (3) more history helps increase the accuracy of forecasting; (4) the PAM is essential for accurate prediction.

## 5 CONCLUSION AND OUTLOOK

In this paper, we propose Pyraformer, a novel model based on pyramidal attention that can effectively describe both short and long temporal dependencies with low time and space complexity. Concretely, we first exploit the CSCM to construct a $C$-ary tree, and then design the PAM to pass messages in both the inter-scale and the intra-scale fashion. By adjusting $C$ and fixing other parameters when the sequence length $L$ increases, Pyraformer can achieve the theoretical $\mathcal{O}(L)$ complexity and $\mathcal{O}(1)$ maximum signal traversing path length. Experimental results show that the proposed model outperforms the state-of-the-art models for both single-step and long-range multi-step prediction tasks, but with less computational time and memory cost. So far we only concentrate on the scenario where $A$ and $S$ are fixed and $C$ increases with $L$ when constructing the pyramidal graph. On the other hand, we have shown in Appendix I that other configurations of the hyper-parameters may further improve the performance of Pyraformer. In the future work, we would like to explore how to adaptively learn the hyper-parameters from the data. Also, it is interesting to extend Pyraformer to other fields, including natural language processing and computer vision.

ACKNOWLEDGEMENT

In this work, Prof. Weiyao Lin was supported by Ant Group through Ant Research Program and in part by National Natural Science Foundation of China under grant U21B2013.

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

Table 4: Meanings of notations.

| Notation | Size | Meaning |
|---|---|---|
| $L$ | Constant | The length of historical sequence. |
| $G$ | Constant | The number of global tokens in ETC. |
| $M$ | Constant | The length of future sequence to be predicted. |
| $B$ | Constant | Batch size. |
| $D$ | Constant | The dimension of each node. |
| $D_K$ | Constant | The dimension of a key. |
| $\boldsymbol{X}$ | $B \times L \times D$ | Input of a single attention head. |
| $\boldsymbol{Y}$ | $B \times L \times D$ | Output of a single attention head. |
| $\boldsymbol{Q}$ | $B \times L \times D_K$ | The query. |
| $\boldsymbol{K}$ | $B \times L \times D_K$ | The key. |
| $\boldsymbol{V}$ | $B \times L \times D_K$ | The value. |
| $\boldsymbol{W}_Q$ | $D \times D_K$ | The weight matrix of the query. |
| $\boldsymbol{W}_K$ | $D \times D_K$ | The weight matrix of the key. |
| $\boldsymbol{W}_V$ | $D \times D_K$ | The weight matrix of the value. |
| $S$ | Constant | Number of scales. |
| $A$ | Constant | Number of adjacent nodes at the same scale that a node can attend to. |
| $C$ | Constant | Number of finer scale nodes that a coarser scale node can summarize. |
| $N$ | Constant | Number of attention layers. |
| $n_l^{(s)}$ | $D$ | The $\ell$-th node at scale s. |
| $\mathbb{N}_\ell^{(s)}$ | $\text{len}(\mathbb{N}_\ell^{(s)}) \times D$ | The set of neighboring nodes of node $n_l^{(s)}$. |
| $\mathbb{A}_\ell^{(s)}$ | $\text{len}(\mathbb{A}_\ell^{(s)}) \times D$ | The adjacent A nodes at the same scale with $n_l^{(s)}$. |
| $\mathbb{C}_\ell^{(s)}$ | $\text{len}(\mathbb{C}_\ell^{(s)}) \times D$ | The children nodes of $n_l^{(s)}$. |
| $\mathbb{P}_\ell^{(s)}$ | $\text{len}(\mathbb{P}_\ell^{(s)}) \times D$ | The parent node of $n_l^{(s)}$. |
| $\boldsymbol{F}_p$ | $B \times M \times D$ | The prediction tokens. |
| $\boldsymbol{F}_e$ | $B \times L_{tot} \times D$ | The output of the encoder. $L_{tot}$ represents the output length of the encoder. |
| $\boldsymbol{F}_{d1}$ | $B \times M \times D$ | The output of the first attention-based decoder layer. |
| $H$ | Constant | The number of attention heads. |
| $D_F$ | Constant | The maximum dimension of the feed-forward layer. |

## A  A BRIEF REVIEW ON RELATED RNN-BASED MODELS

In this section, we provide a brief review on the related RNN-based models. Multiscale temporal dependencies are successfully captured in HRNN (Costa-jussà & Fonollosa, 2016) and HM-RNN (Chung et al., 2019). The former requires expert knowledge to partition the sequence into different resolutions, while the latter learns the partition automatically from the data. Note that the theoretical maximum length of the signal traversing path in both models is still $\mathcal{O}(L)$. Another line of works aim to shorten the signal traversing path by adding residual connections (Kim et al., 2017) or dilated connections to LSTMs (Chang et al., 2017). However, they do not consider the multiresolution temporal dependencies explicitly. Furthermore, all aforementioned RNNs only propagate information in one direction from the past to the future. An appealing approach that allows bidirectional information exchange is Bi-LSTM (Schuster, 1996). The forward and backward propagation is realized through two different LSTMs though, and so still incurs a long signal traversing path.

As opposed to the abovementioned RNN-based models, the proposed Pyraformer enables bidirectional information exchange that can better describe the temporal dependencies, while providing a multiresolution representation of the observed sequence at the same time. We also notice that due to the unidirectional property of RNNs, it is difficult the realize the pyramidal graph in Figure 1d based on RNNs.

## B  PROOF OF LEMMA 1

*Proof.* Let $S$ denote the number of scales in the pyramidal graph, $C$ the number of children nodes in the finer scale $s-1$ that a node in the the coarser scale $s$ can summarize for $s = 2, \cdots, S$, $A$ the number of adjacent nodes that a node can attend to within each scale, $N$ the number of attention layers, and $L$ the length of the input time series. We define the term "receptive field" of an arbitrary node $n_a$ in a graph as the set of nodes that $n_a$ can receive messages from. We further define the distance between two arbitrary nodes in a graph as the length of the shortest path between them (i.e., the number of steps to travel from one node to another). Note that in each attention layer, the messages can only travel by one step in the graph.

Without sacrificing generality, we assume that $L$ is divisible by $C^{S-1}$, and then the number of nodes at the coarsest scale $S$ is $L/C^{S-1}$. Since every node is connected to $A$ closest nodes at the same scale, the distance between the leftmost and the rightmost node at the coarsest scale is $2(L/C^{S-1} - 1)/(A - 1)$. Hence, the leftmost and the rightmost node in the coarsest scale are in the receptive field of each other after the stack of $N \geq 2(L/C^{S-1} - 1)/(A - 1)$ layers of the pyramidal attention. In addition, owing to the CSCM, nodes at the coarsest scale can be regarded as the summary of the nodes in the finer scales. As a result, when Equation (4) is satisfied, all nodes at the coarsest scale have a global receptive field, which closes the proof. □

## C  PROOF OF PROPOSITION 1

*Proof.* Suppose that $L^{(s)}$ denotes the number of nodes at scale $s$, that is,

$$L^{(s)} = \frac{L}{C^{s-1}}, 1 \leq s \leq S. \tag{6}$$

For a node $n_\ell^{(s)}$ in the pyramidal graph, the number of dot products $P_\ell^{(s)}$ it acts as the query can be decomposed into two parts:

$$P_\ell^{(s)} = P_\ell^{(s)}{}_{\text{inter}} + P_\ell^{(s)}{}_{\text{intra}}, \tag{7}$$

where $P_\ell^{(s)}{}_{\text{intra}}$ and $P_\ell^{(s)}{}_{\text{inter}}$ denotes the intra-scale and the inter-scale part respectively. According to the structure of the pyramidal graph, we can have the following inequalities:

$$P_\ell^{(s)}{}_{\text{intra}} \leq A, \tag{8}$$

$$P_\ell^{(s)}{}_{\text{inter}} \leq C + 1. \tag{9}$$

The first inequality (8) holds since a node typically attends to $A$ most adjacent nodes at the same scale but for the leftmost and the rightmost node, the number of in-scale nodes it can attend to is smaller than $A$. On the other hand, the second inequality (9) holds because a node typically has $C$ children and 1 parent in the pyramidal graph but nodes at the top and the bottom scale can only attend to fewer than $C + 1$ nodes at adjacent scales.

In summary, the number of dot products that need to be calculated for scale $s$ is:

$$P^{(s)} = \sum_{\ell=1}^{L^{(s)}} \left( P_\ell^{(s)}{}_{\text{intra}} + P_\ell^{(s)}{}_{\text{inter}} \right) \leq L^{(s)}(A + C + 1). \tag{10}$$

Note that $P^{(1)} \leq L(A + 1)$ for the finest scale (i.e., $s = 1$) since nodes at this scale do not have any children. It follows that the number of dot products that need to be calculated for the entire pyramidal attention layer is:

$$P = \sum_{s=1}^{S} P^{(s)}$$

$$\leq L(A+1) + L^{(2)}(A+C+1) + ... + L^{(S)}(A+C+1)$$

$$= L\left(\sum_{s=1}^{S} C^{-(s-1)}A + \sum_{s=2}^{S} C^{-(s-1)} + \sum_{s=1}^{S-1} C^{-(s-1)} + 1\right)$$

$$< L\left((A+2)\sum_{s=1}^{S} C^{-(s-1)} + 1\right). \tag{11}$$

In order to guarantee that the nodes at the coarsest scale have a global receptive field, we choose $C$ such that $C \propto {}^{S-1}\!\sqrt{L}$. Consequently, the complexity of the proposed pyramidal attention is:

$$\mathcal{O}(P) \leq \mathcal{O}\left(L\left((A+2)\sum_{s=1}^{S} C^{-(s-1)} + 1\right)\right)$$

$$= \mathcal{O}\left(L(A+2)\sum_{s=1}^{S} C^{-(s-1)}\right)$$

$$= \mathcal{O}\left(\frac{(A+2)L^{\frac{S}{S-1}} - 1}{L^{\frac{1}{S-1}} - 1}\right)$$

$$= \mathcal{O}\left(\frac{AL^{\frac{S}{S-1}} - 1}{L^{\frac{1}{S-1}} - 1}\right). \tag{12}$$

When $L$ approaches infinity, the above expression amounts to $\mathcal{O}(AL)$. Since $A$ can be fixed when $L$ changes, the complexity can be further reduced to $\mathcal{O}(L)$. □

## D   PROOF OF PROPOSITION 2

*Proof.* Let $n_\ell^{(s)}$ represent the $\ell$-th node of the $s$-th scale. It is evident that the distance between $n_1^{(1)}$ and $n_L^{(1)}$ is the largest among all pairs of nodes in the pyramidal graph. The shortest path to travel from $n_1^{(1)}$ to $n_L^{(s)}$ is:

$$n_1^{(1)} \to n_1^{(2)} \to \cdots \to n_1^{(S)} \to \cdots \to n_{L^{(S)}}^{(S)} \to n_{L^{(S-1)}}^{(S-1)} \to \cdots \to n_L^{(1)}. \tag{13}$$

Correspondingly, the length of the maximum path between two arbitrary nodes in the graph is:

$$L_{\max} = 2(S-1) + \frac{2(L^{(S)} - 1)}{A - 1}. \tag{14}$$

When $C$ satisfies Equation (5), that is, $L^{(S)} - 1 \leq (A-1)N/2$, we can obtain:

$$\mathcal{O}(L_{\max}) = \mathcal{O}\left(2(S-1) + \frac{2(L^{(S)} - 1)}{A - 1}\right)$$

$$= \mathcal{O}\left(2(S-1) + \frac{2(\frac{L}{C^{S-1}} - 1)}{A - 1}\right)$$

$$= \mathcal{O}(2(S-1) + N)$$

$$= \mathcal{O}(S + N). \tag{15}$$

Since $A$, $S$ and $N$ are invariant with $L$, the order of the maximum path length $L_{\max}$ can be further simplified as $\mathcal{O}(1)$. □

## E   DATASETS

We demonstrated the advantages of the proposed Pyraformer on the following four datasets. The first three datasets were used for single-step forecasting, while the last two for long-range multi-step forecasting.

**Wind**[2]: This dataset contains hourly estimation of the energy potential in 28 countries between 1986 and 2015 as a percentage of a power plant's maximum output. Compared with the remaining datasets, it is more sparse and periodically exhibits a large number of zeros. Due to the large size of this dataset, the ratio between training and testing set was roughly 32:1.

**App Flow**: This dataset was collected at Ant Group[3]. It consists of hourly maximum traffic flow for 128 systems deployed on 16 logic data centers, resulting in 1083 different time series in total. The length of each series is more than 4 months. Each time series was divided into two segments for training and testing respectively, with a ratio of 32:1.

**Electricity**[4] (Yu et al., 2016): This dataset contains time series of electricity consumption recorded every 15 minutes from 370 users. Following DeepAR (Salinas et al., 2020), we aggregated every 4 records to get the hourly observations. This dataset was employed for both single-step and long-range forecasting. We trained with data from 2011-01-01 to 2014-09-01 for single-step forecasting, and from 2011-04-01 to 2014-04-01 for long-range forecasting.

**ETT**[5] (Zhou et al., 2021): This dataset comprises 2 years of 2 electricity transformers collected from 2 stations, including the oil temperature and 6 power load features. Observations every hour (i.e., ETTh1) and every 15 minutes (i.e., ETTm1) are provided. This dataset is typically exploited for model assessment on long-range forecasting. Here, we followed Informer (Zhou et al., 2021) and partitioned the data into 12 and 4 months for training and testing respectively.

## F    EXPERIMENT SETUP

We set $S = 4$ and $N = 4$ for Pyraformer in all experiments. When the historical length $L$ is not divisible by $C$, we only introduced $\lfloor L/C \rfloor$ nodes in the upper scale, where $\lfloor \cdot \rfloor$ denotes the round down operation. The last $L - (\lfloor L/C \rfloor - 1)C$ nodes at the bottom scale were all connected to the last node at the upper scale. For single-step forecasting, we set $C = 4$, $A = 3$, and $H = 4$ in all experiments. Both training and testing used a fixed-size historical sequence to predict the mean and variance of the Gaussian distribution of a single future value. We chose the MSE loss and the log-likelihood (Zuo et al., 2020) as our loss functions. The ratio between them was set to 100. For optimization, we used Adam with the learning rate starting from $10^{-5}$ and halving in every epoch. We trained Pyraformer with 10 epochs. Weighted sampler based on each window's average value and hard sample mining were used to improve the generalization ability of the network. On the other hand, for long-range forecasting, we tested four combinations of $A$ and $C$ in each experiment, and the best results were presented. Specifically, when the prediction length is smaller than 600, we tested $A = 3, 5$ and $C = 4, 5$. When the prediction length is larger than 600, we tested $A = 3, 5$ and $C = 5, 6$. The resulting choice of hyper-parameters for each experiment is listed in Table 5. In addition, the loss function was the MSE loss only. We still used Adam as our optimizer, but the learning rate started from $10^{-4}$ and was reduced to one-tenth every epoch. We set the number of epochs to be 5.

## G    PRETRAINING

For single-step forecasting, the value to be predicted is usually close to the last value of history. Since we only use the last nodes of all scales to predict, the network tends to focus only on short-term dependencies. To force the network to capture long-range dependencies, we add additional supervision in the first few epochs of training. Specifically, in the first epoch, we form our network as an auto-encoder, as shown in Figure 5. Apart from predicting future values, the PAM is also

---

[2]Wind dataset can be downloaded at https://www.kaggle.com/sohier/30-years-of-european-wind-generation

[3]The App Flow dataset does not contain any Personal Identifiable Information and is desensitized and encrypted. Adequate data protection was carried out during the experiment to prevent the risk of data copy leakage, and the dataset was destroyed after the experiment. It is only used for academic research, it does not represent any real business situation. The download link is https://github.com/alipay/Pyraformer/tree/master/data/app_zone_rpc_hour_encrypted.csv

[4]Electricity dataset can be downloaded at https://archive.ics.uci.edu/ml/datasets/ElectricityLoadDiagrams20112014

[5]ETT dataset can be downloaded at https:// github.com/zhouhaoyi/ETDataset

Table 5: Hyper-parameter settings of long-range experiments.

| Dataset | prediction length | N | S | H | A | C | historical length |
|---------|------------------|---|---|---|---|---|-------------------|
| ETTh1 | 168 | 4 | 4 | 6 | 3 | 4 | 168 |
| | 336 | 4 | 4 | 6 | 3 | 4 | 168 |
| | 720 | 4 | 4 | 6 | 5 | 4 | 336 |
| ETTm1 | 96 | 4 | 4 | 6 | 3 | 5 | 384 |
| | 288 | 4 | 4 | 6 | 5 | 5 | 672 |
| | 672 | 4 | 4 | 6 | 3 | 6 | 672 |
| Elect | 168 | 4 | 4 | 6 | 3 | 4 | 168 |
| | 336 | 4 | 4 | 6 | 3 | 4 | 168 |
| | 720 | 4 | 4 | 6 | 3 | 5 | 336 |

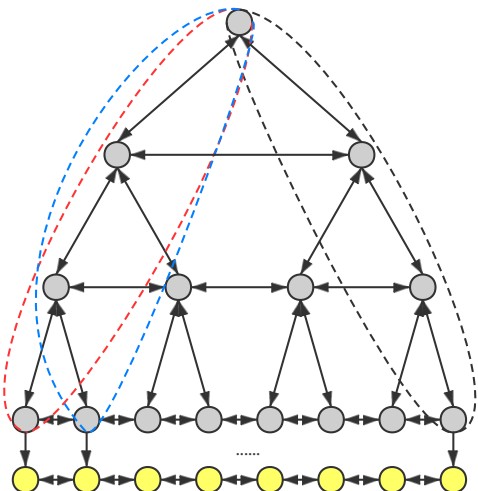

Figure 5: The pretraining strategy for one-step prediction. Features of nodes surrounded by the dashed ellipses are concatenated to recover the corresponding input value.

trained to recover the input values. Note that we test all methods with and without this pretraining strategy and the better results are displayed in Table 2.

## H  METRICS

Denote the target value as $z_{j,t}$ and the predicted value as $\hat{z}_{j,t}$, where $j$ is the sample index and $t$ is the time index. Then NRMSE and ND are calculated as follows:

$$\text{NRMSE} = \frac{\sqrt{\frac{1}{NT}\sum_{j=1}^{N}\sum_{t=1}^{T}(z_{j,t}-\hat{z}_{j,t})^2}}{\frac{1}{NT}\sum_{j=1}^{N}\sum_{t=1}^{T}|z_{j,t}|}, \tag{16}$$

$$\text{ND} = \frac{\sum_{j=1}^{N}\sum_{t=1}^{T}|z_{j,t}-\hat{z}_{j,t}|}{\sum_{j=1}^{N}\sum_{t=1}^{T}|z_{j,t}|}. \tag{17}$$

## I  EXPERIMENTS ON SYNTHETIC DATA

To further evaluate Pyraformer's ability to capture different ranges of temporal dependencies, we synthesized an hourly dataset with multi-range dependencies and carried out experiments on it.

Specifically, each time series in the synthetic dataset is a linear combination of three sine functions of different periods: 24, 168 and 720, that is,

$$f(t) = \beta_0 + \beta_1 \sin(\frac{2\pi}{24}t) + \beta_2 \sin(\frac{2\pi}{168}t) + \beta_3 \sin(\frac{2\pi}{720}t). \tag{18}$$

In the above equation, the coefficients of the three sine functions $\beta_1$, $\beta_2$, and $\beta_3$ for each time series are uniformly sampled from $[5, 10]$. $\beta_0$ is a Gaussian process with a covariance function $\Sigma_{t_1,t_2} = |t_1 - t_2|^{-1}$ and $\Sigma_{t_1} = \Sigma_{t_2} = 1$, where $t_1$ and $t_2$ denote two arbitrary time stamps. Such polynomially decaying covariance functions are known to have long-range dependence, as oppose to the exponentially decaying covariance functions (Yu et al., 2019). The start time of each time series $t_0$ is uniformly sampled from $[0, 719]$. We first generate 60 time series of length 14400, and then split each time series into sliding windows of width 1440 with a stride of 24. In our experiments, we use the historical 720 time points to predict the future 720 points. Since both the deterministic and stochastic parts of the synthetic time series have long-range correlations, such dependencies should be well captured in the model in order to yield accurate predictions of the next 720 points. The results are summarized in Table 6. Here, we consider two different configurations of Pyraformer: 1) $C = 6$ for all scales in the pyramidal graph (denoted as Pyraformer$_{6,6,6}$); 2) $C = 12$, $7$, and $4$ for the three layers sequentially from bottom to top (denoted as Pyraformer$_{12,7,4}$).

Table 6: Long-range forecasting results on the synthetic dataset.

| Method | MSE | MAE |
|---|---|---|
| Full attention | 3.550 | 1.477 |
| LogTrans | 3.007 | 1.366 |
| ETC | 4.742 | 5.509 |
| Informer | 7.546 | 2.092 |
| Longformer | 2.032 | 1.116 |
| Reformer | 1.538 | 3.069 |
| Pyraformer$_{6,6,6}$ | 1.258 | 0.877 |
| Pyraformer$_{12,7,4}$ | 1.176 | 0.849 |

It can be observed that Pyraformer$_{6,6,6}$ with the same $C$ for all scales already outperforms the benchmark methods by a large margin. In particular, the MSE given by Pyraformer is decreased by $18.2\%$ compared with Reformer, which produces the smallest MSE among the existing variants of Transformer. On the other hand, by exploiting the information of the known period, Pyraformer$_{12,7,4}$ performs even better than Pyraformer$_{6,6,6}$. Note that in Pyraformer$_{12,7,4}$, nodes at scale 2, 3, and 4 characterizes coarser temporal resolutions respectively corresponding to half a day, half a week, and half a month. We also tested Pyraformer$_{24,7,4}$, but setting $C = 24$ in the second scale degrades the performance, probably because the convolution layer with a kernel size of 24 is difficult to train.

We further visualized the forecasting results produced by Pyraformer$_{12,7,4}$ in Figure 6. The blue solid curve and red dashed curve denote the true and predicted time series respectively. By capturing the temporal dependencies with different ranges, the prediction resulting from Pyraformer closely follows the ground truth.

On the other hand, to check whether Pyraformer can extract features with different temporal resolutions, we depicted the extracted features in a randomly selected channel across time at each scale in the pyramidal graph in Figure 7. It is apparent that the features at the coarser scales can be regarded as a lower resolution version of the features at the finer scales.

## J  ABLATION STUDY

### J.1  IMPACT OF $A$ AND $C$

We studied the impact of $A$ and $C$ on the performance of Pyraformer for long-range time series forecasting, and showed the results in Table 7. Here, we focus on the dataset ETTh1. The history

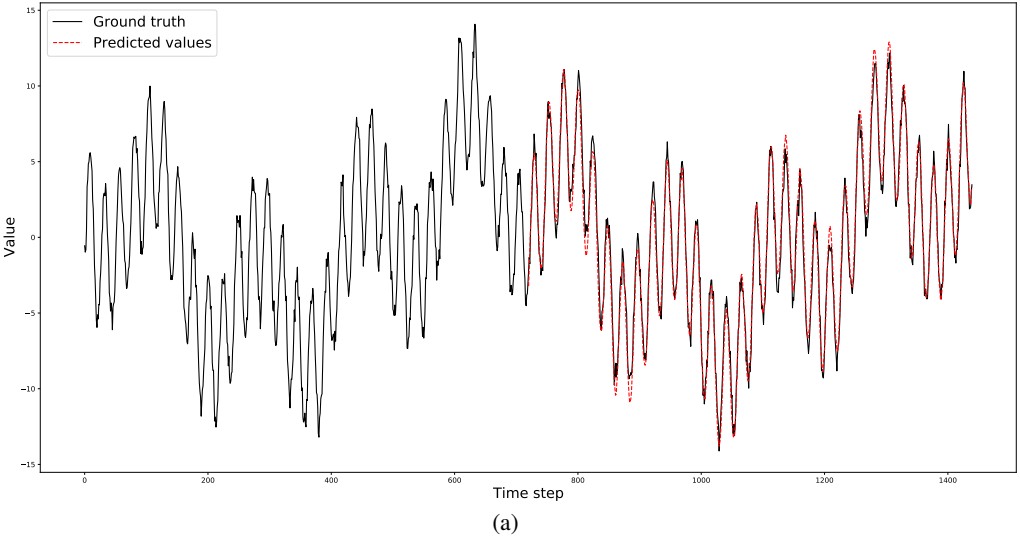

Figure 6: Visualization of prediction results on the synthetic dataset.

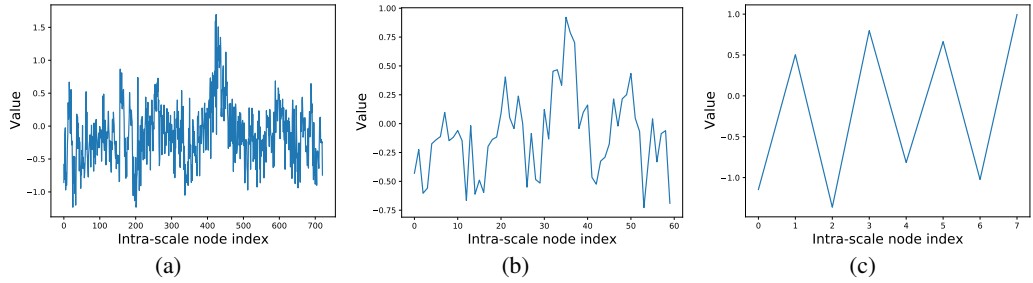

Figure 7: Visualization of the extracted features across time in second channel at different scales: (a) scale 1; (b) scale 2; (c) scale 3.

length is 336 and the prediction length is 720. From Table 7, we can conclude that the receptive fields of the nodes at the coarsest scale in the PAM play an indispensable role in reducing the prediction error of Pyraformer. For instance, there are $42$ nodes at the coarsest scale when $C = 2$. Without the intra-scale connections, each node can only receive messages from $16$ nodes at the finest scale. As the number of adjacent connections $A$ in each scale increases, the receptive fields of the coarsest-scale nodes also extend, and therefore, the prediction error decreases accordingly. However, as long as the nodes at the top scale have a global receptive field, further increasing $A$ will not bring large gains. For $C = 5$, the performance does not improve even though $A$ increases. Such observations indicate that it is better to set $A$ to be small once the uppermost nodes in the PAM have a global receptive field. In practice, we only increase $C$ with the increase of $L$, but keep $A$ small.

## J.2 IMPACT OF THE CSCM ARCHITECTURE

In addition to convolution, there exist other mechanisms for constructing the $C$-ary tree, such as max pooling and average pooling. We studied the impact of different CSCM architectures on the performance for long-range forecasting on dataset ETTh1. The history and the prediction length are both 168 and $C = 4$ for all mechanisms. The results are listed in Table 8. From Table 8, we can tell that: (1) Using pooling layers instead of convolution typically degrades the performance. However, the performance of Pyraformer based on max pooling is still superior to that of Informer, demonstrating the advantages of the PAM over the prob-sparse attention in Informer. (2) The MSE of convolution with the bottleneck is only $1.51\%$ larger than that without bottleneck, but the number

Table 7: Impact of $A$ and $C$ on long-range forecasting. The history length is 336.

| | $A = 3$ | | | $A = 9$ | | | $A = 13$ | | |
| --- | --- | --- | --- | --- | --- | --- | --- | --- | --- |
| | MSE | MAE | Q-K pairs | MSE | MAE | Q-K pairs | MSE | MAE | Q-K pairs |
| $C = 2$ | 1.035 | 0.811 | 73512 | 1.029 | 0.815 | 162648 | 1.003 | 0.807 | 221112 |
| $C = 3$ | 1.029 | 0.817 | 58992 | 1.009 | 0.798 | 128976 | 1.056 | 0.805 | 174672 |
| $C = 4$ | 1.001 | 0.802 | 53208 | 1.028 | 0.806 | 115848 | 1.027 | 0.804 | 156696 |
| $C = 5$ | **0.999** | **0.796** | **49992** | 1.005 | 0.796 | 108744 | 1.017 | 0.797 | 147192 |

Table 8: Impact of the CSCM architecture on long-range forecasting. Parameters introduced by the normalization layers are relatively few, and thus, are ignored.

| CSCM | MSE | MAE | Parameters |
| --- | --- | --- | --- |
| Max-pooling | 0.842 | 0.700 | 0 |
| Average-pooling | 0.833 | 0.693 | 0 |
| Conv. | **0.796** | **0.679** | 3147264 |
| Conv. w/bottleneck | 0.808 | 0.683 | 328704 |

Table 9: Impact of history length. The prediction length is 1344.

| History Length | MSE | MAE |
| --- | --- | --- |
| 84 | 1.234 | 0.856 |
| 168 | 1.226 | 0.868 |
| 336 | 1.108 | 0.835 |
| 672 | 1.057 | 0.806 |
| 1344 | 1.062 | 0.806 |

Table 10: Impact of the PAM.

| Method | Metrics | 96 | 288 | 672 |
| --- | --- | --- | --- | --- |
| CSCM Only | MSE | 0.576 | 0.782 | 0.883 |
| | MAE | 0.544 | 0.683 | 0.752 |
| Pyraformer | MSE | **0.480** | **0.754** | **0.857** |
| | MAE | **0.486** | **0.659** | **0.707** |

of parameters is reduced by almost 90%. Thus, we adopt the more compact module of convolution with bottleneck as our CSCM.

### J.3 IMPACT OF THE HISTORY LENGTH

We also checked the influence of the history length on the prediction accuracy. The dataset is ETTm1, since its granularity is minute and contains more long-range dependencies. We fixed the prediction length to 1344 and changed the history length from 84 to 1344 in Table 9. As expected, a longer history typically improves prediction accuracy. On the other hand, this performance gain starts to level off when introducing more history stops providing new information. As shown in Figure 8, the time series with length 672 contains almost all periodicity information that is essential for prediction, while length 1344 introduces more noise.

### J.4 IMPACT OF THE PAM

Finally, we investigated the importance of the PAM. We compared the performance of Pyraformer with and without the PAM on the dataset ETTm1. For a fair comparison, the number of parameters of the two methods were controlled to be within the same order of magnitude. More precisely, we increased the bottleneck dimension of "Conv. w/bottleneck" for the model only with the CSCM. The results are shown in Table 10. Obviously, the PAM is vital to yield accurate predictions.

## K DISCUSSION ON THE SELECTION OF HYPER-PARAMETERS

We recommend to first determine the number of attention layers $N$ based on the available computing resources, as this number is directly related to the model size. Next, the number of scales $S$ can be determined by the granularity of the time series. For example, for hourly observations, we typically assume that it may also have daily, weekly and monthly periods. Therefore, we can set $S$ to be 4. We then focus on the selection of $A$ and $C$. According to the ablation study, we typically prefer a small $A$, such as 3 and 5. Lastly, in order to ensure the network has a receptive field of $L$, we can

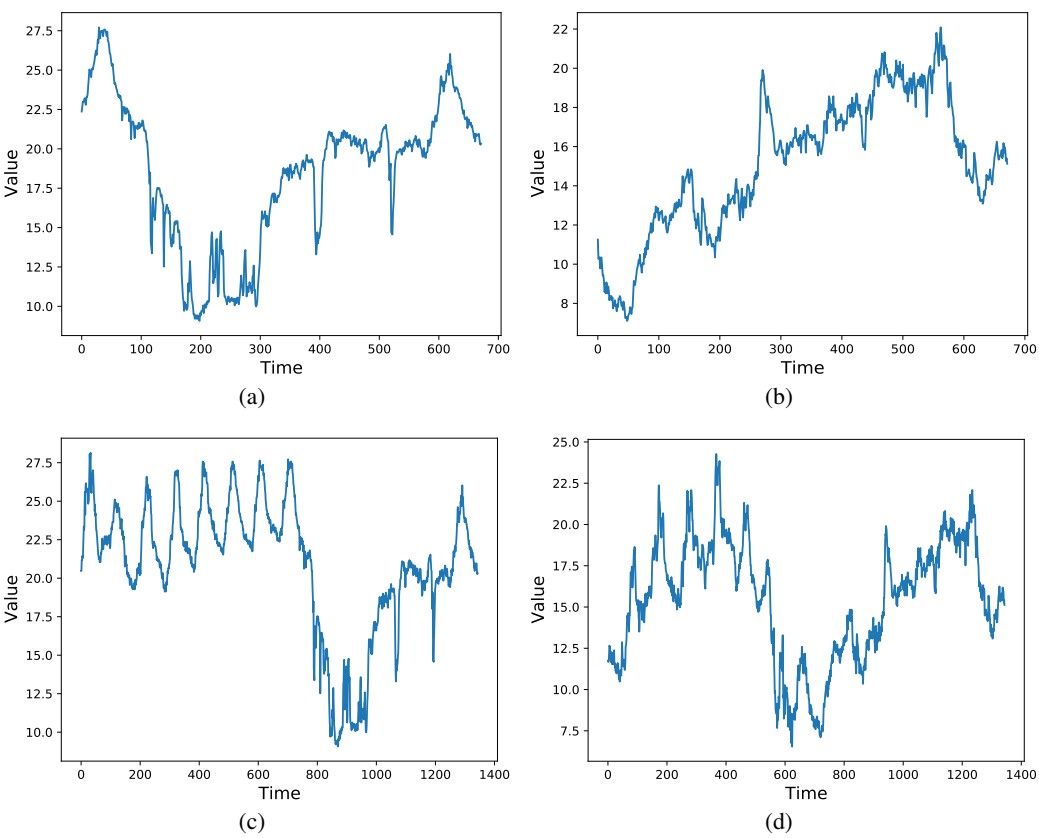

Figure 8: Time series with different lengths in the ETTm1 dataset. The sequence length in (a) and (b) is 672, and that in (c) and (d) is 1344. The time series in (a) and (b) corresponds to the latter half of those in (c) and (d) respectively.

select a $C$ that satisfies Equation (5). In practice, we can use a validation set to choose $C$ from its candidates that satisfies (5). It is also worthwhile to check whether choosing different $C$ for different scales based on the granularity of the time series can further improve the performance as we did in Appendix I.

