# OpenReview forum: "Pyraformer: Low-Complexity Pyramidal Attention for Long-Range Time Series Modeling and Forecasting"
_ICLR.cc/2022/Conference — ICLR 2022 Oral_

### Official Review · Reviewer_17as · 2021-10-29

**Correctness:** 4
**Technical Novelty And Significance:** 3
**Empirical Novelty And Significance:** 3
**Recommendation:** 8
**Confidence:** 3

**Main Review:**

__Pros__

1. Overall I like the idea conceptually. As mentioned in the summary it makes a lot of sense for time-series data as it has an implicit bias of summarizing the past at different resolutions.

2. This architecture as claimed, can handle long sequences with computational complexity O(L) where $L$ is the number of elements in the sequence. It can also maintain short attention path distance between any node. This distance is O(1) w.r.t L.

3. The experiments are good enough to demonstrate the advantage of this network over other attention mechanisms.

4. The implementation has been done efficiently using TVM and when released the code base would be of value. I hope the paper is reproducible even though no code has been provided.

5. The empirical speed and memory consumption graphs are useful.

__Cons__

I have certain clarifying questions. i would be happy to raise my scores if these are answered. Further I have some comments that would hopefully make the paper better.

1. It would be better to formally define Signal Traversing Path and forward reference it from the introduction.

2. While reading the initial description in the introduction, it is not fully clear what the nodes at each resolution represent. This is only evident to me after reading section 3.2 which states that convolutional layers with specific strides are applied repeatedly to get the coarser levels. It would be better if one states upfront that some external mechanism is used to extract the initial states of the coarser nodes. Related comment: I think the clarity of section 3,2 can be improved.

3. I might be wrong but I believe N is not defined in section 3.1 before it is used. Please check this (ignore if I missed this somehow).What is the relation between N and S ? I would suggest defining A, L, C, N, S clearly in one place with some indentation in Section 3.1, for ease of reading.

4. Please be slightly more clear in the second paragraph of section 3. 3. I would like to verify my understanding of the multi-step forecasting modules. In the first module, we just have a fixed output layer which is a dense mapping to F outputs, where F is the future prediction length. In the second mapping, there is a decoder that sequentially generated the multi-step output and the decoder is equivalent to the decoder in the original transformer paper. Is my understanding correct?

5. In the experiments the information of the datasets and the experimental setup is not 100% clear to me even after reading the appendix. Please clarify or point me to the relevant places in the paper. In multi-step forecasting what is the length of the future prediction window. Also, is the task rolling prediction over the test set? These should be clearly defined.

6. is there a validation set? Typically I would prefer if A and C are tuned on a validation set per dataset and the best results are reported. the ordering would stay the same I guess, because fixed set of A, C are used in all experiments.

7. Is there a relation between A, C vs the granularity of the dataset. For example if the data is hourly, then one could build a resolution graph such that groups of 24 are constructed in the first level (24 hours in a day), then groups of 7 (7 days in a week), then 4 weeks in a month. Such a non-uniform pyramidal structure might be very useful. Can the authors comment on this or better yet try this? -- not absolutely essential but it would be interesting.

8. Generally, it would be better if we could rank pyramidal attention vs state of the art models on the well known benchmarks used in the DCRNN paper and newer papers like https://arxiv.org/abs/2103.07719. For instance one could just do the evaluation of pyramidal attention in the same tasks as the linked paper and report the numbers. This would reveal the rank of pyramidal attention in the SOTA table and would be a useful signal for the community. Again this is not required but a nice to have.




**Summary Of The Paper:**

The paper proposes a variant of the transformer architecture called pyramidal attention. In this architecture one forms a pyramid graph over the sequence at different resolutions and then applies attention among the neighbors of that graph. This leads to a network that can attend to long sequences of length L with O(L) computation but at the same time attention path between two sequence positions is O(1). This is especially relevant for time-series forecasting as it has potential to summarize the time-series at different scales like hourly, daily, monthly etc. Some experiments are performed on 4 datasets to show the advantage of this architecture over other transformer variants.

**Summary Of The Review:**

I like the overall idea and the execution is decent (please see the pros and cons above). In my mind the pros outweigh the cons and I am currently rating it as above the acceptance threshold. I can strengthen my reviews if the authors answer the questions asked in the cons section.

---

> ### Author Response · Authors · 2021-11-22
> **Response to reviewer 17as, part3**
>
> > *7. Is there a relation between A, C vs the granularity of the dataset? For example, if the data is hourly, then one could build a resolution graph such that groups of 24 are constructed in the first level (24 hours in a day), then groups of 7 (7 days in a week), then 4 weeks in a month. Such an irregular pyramidal structure might be very useful.*
>
> We fully acknowledge that setting $A$ and $C$ based on the granularity of the dataset is an interesting idea, and we try to explore the potential of the non-uniform pyramidal graph with different $C$ for each scale with a synthetic dataset in **Appendix I**.
>
> Specifically, each time series in the synthetic dataset is a linear combination of three sine functions of different periods: 24, 168 and 720, that is,
>
> \begin{equation}
>     f(t)= \beta_0 + \beta_1 \sin(\frac{2\pi}{24}t) + \beta_2 \sin(\frac{2\pi}{168}t) + \beta_3 \sin(\frac{2\pi}{720}t).
> \end{equation}
>
> In the above equation, the coefficients of the three sine functions $\beta_1$, $\beta_2$, and $\beta_3$ for each time series are uniformly sampled from $[5, 10]$. $\beta_0$ is a Gaussian process with a covariance function $\Sigma_{t_1, t_2} = |t_1 - t_2|^{-1}$ and $\Sigma_{t_1} = \Sigma_{t_2} = 1$, where $t_1$ and $t_2$ denote two arbitrary time stamps. Such polynomially decaying covariance functions are known to have long-range dependence [2]. We first generate 60 time series of length 14400, and then split each time series into sliding windows of width 1440 with a stride of 24. In our experiments, we use the historical 720 time points to predict the future 720 points. With this long-range correlated noise and sine functions of different periods, the model needs to be able to capture multi-resolution temporal dependencies to accurately predict the next 720 points. The results are summarized below. The subscripts of Pyraformer correspond to the $C$ in each scale from bottom to top.
>
> | Method         |    MSE   |   MAE   |
> | ----------------- | ---------- | ---------- |
> | Full attention|    3.550  |   1.477  |
> | LogTrans     |   3.007   |   1.366  |
> | ETC             |   4.742   |   5.509  |
> | Informer       |    7.546  |    2.092 |
> | Longformer  |    2.032 | 1.116 |
> | Reformer     | 1.538 | 3.069 |
> | Pyraformer-[6,6,6]   | 1.258 | 0.877 |
> | Pyraformer-[24,7,4] | 1.776 | 1.033 |
> | Pyraformer-[12,7,4] | 1.176 | 0.849 |
>
> It can be observed that under the regular pyramidal structure (choose $C=6$ for all scales), Pyraformer already outperforms other methods by a large margin. Moreover, we set different $C$ for different scales, and find that Pyraformer-[12,7,4] performs even better than that of Pyraformer-[6, 6, 6]. In this setting, nodes at scale 2, 3 and 4 represents half a day, half a week, and half a month, respectively, and the pyramidal attention successfully utilizes prior period knowledge to further improve the prediction accuracy. On the other hand, we find that the performance of Pyraformer-[24,7,4] is not as good as Pyraformer-[6, 6, 6}], probably because the convolution layer with a kernel size of 24 is difficult to train.
>
> We have also added visualizations of multi-resolution features in **Appendix I**. It can be observed that the coarser-scale features are approximately a summary of the finer-scale features.
>
> > *8. Generally, it would be better if we could rank pyramidal attention vs state of the art models on the well known benchmarks used in the DCRNN paper and newer papers like https://arxiv.org/abs/2103.07719. For instance one could just do the evaluation of pyramidal attention in the same tasks as the linked paper and report the numbers.*
>
> We have checked DCRNN and the paper in the link carefully, and noticed that both of them focus on capturing the interactions between different time series under the scenario of multivariate time series forecasting. Moreover, the historical length in this two models is typically very short. As a result, we think it is difficult to apply Pyraformer in this scenario, since the proposed model typically requires a long history to extract the long-range dependencies and is devoted to modeling the complicated temporal dependencies within a time series instead of the spatial dependencies between different time series. Due to the limited discussion time, we decide not to explore the potential use of the pyramidal attention in this direction.
>
> ### Reference
>
> [1] Haoyi Zhou, Shanghang Zhang, Jieqi Peng, Shuai Zhang, Jianxin Li, Hui Xiong, and Wancai Zhang. "Informer: Beyond efficient transformer for long sequence time-series forecasting." In Proceedings of AAAI, 2021.
>
> [2] Choi, Myung Jin, Venkat Chandrasekaran, and Alan S. Willsky. "Exploiting sparse Markov and co-variance structure in multiresolution models." In Proceedings of the 26th Annual International Conference on Machine Learning.

---

> > ### Comment · Reviewer_17as · 2021-11-29
> > **Thanks for the response**
> >
> > I have upgraded my score to 8. I think the rebuttal addresses most of my concerns.

---

> > > ### Author Response · Authors · 2021-11-30
> > > **Thanks for the Response of Reviewer 17as**
> > >
> > > Many thanks to Reviewer 17as for providing an impressively insightful pre-rebuttal review. Your detailed suggestions help us a lot in paper revision.
> > >
> > > We'd also thank your dedication for carefully judging our feedback and raising the score! Your constructive suggestions are very helpful for us to improve the paper in a better shape.

---

> ### Author Response · Authors · 2021-11-22
> **Response to reviewer 17as, part2**
>
> ''For multi-step forecasting, we propose two prediction modules. The first one is the same with the single-step forecasting module, but **it maps the last nodes at all scales to all $M$ future time steps in a batch**. The second one, on the other hand, resorts to a decoder with two full attention layers. Specifically, similar to the original Transformer (Vaswani et al., 2017), we replace the observations at the future $M$ time steps with 0, embed them in the same manner with the historical observations, and refer to the summation of the observation, covariate, and positional embedding as the ''prediction token'' $\mathbf{F_p}$. The first attention layer then takes the prediction tokens $\mathbf{F_p}$ as the query and the output of the encoder $\mathbf{F_e}$ (i.e., all nodes in the PAM) as the key and the value, and yields $\mathbf{F_{d1}}$. The second layer takes $\mathbf{F_{d1}}$ as the query, but takes the concatenated $\mathbf{F_{d1}}$ and $\mathbf{F_e}$ as the key and the value. The historical information $\mathbf{F_e}$ is fed directly into both attention layers, since such information is vital for accurate long-range forecasting. The final prediction is then obtained through a fully connected layer across the dimension of channels. **Again, we output all future predictions together to avoid the problem of error accumulation in the autoregressive decoder of Transformer.**''
>
> > *5. Clarify the information of the datasets and the experimental setup. In multi-step forecasting what is the length of the future prediction window. Also, is the task rolling prediction over the test set?*
>
> We have added relevant statements in **Appendix F**. In multi-step forecasting, the prediction length is shown in Table 3, and we have added the corresponding historical length for each prediction length in **Appendix F**.
>
> For the prediction on the test set, we follow the scheme in Informer [1]. More concretely, we partition the testing data into sliding windows with stride 24. The width of each window equals the historical length plus the prediction length. We then use the model trained on the training data and forecast the prediction part in each window given the history part.
>
> > *6. Is there a validation set?*
>
> We agree that it is better to use a validation set to tune the hyper-parameters. We have included a discussion on how to choose the hyper-parameters in **Appendix K**.
>
> For the sake of simplicity, we did not use a validation set in our experiments. Instead, for single-step forecasting, we fixed $A = 3$ and $C = 4$. On the other hand, for long-range multi-step forecasting, we tested four combinations of $A$ and $C$ in each experiment, and the best results were selected. Specifically, when the prediction length is smaller than 600, we tested $A = 3,5$ and $C = 4,5$. When the prediction length is larger than 600, we tested $A = 3,5$ and $C = 5,6$. We have made this point clear in **Appendix F**.
>
> In addition, we have also followed your suggestion and partitioned the ETTh1 dataset into a training and a validation set, and then tuned $A$ and $C$ on the validation set. As expected, the order stays the same.

---

> ### Author Response · Authors · 2021-11-22
> **Response to reviewer 17as, part1**
>
> We appreciate your thoughtful and insightful comments. We believe that addressing the reviewer's comments has resulted in improving the clarity and presentation of the paper’s contributions and has brought the paper to a higher standard. Below we address your concerns point by point. Corresponding modifications in the paper are **highlighted in blue**. Unless otherwise stated in our response, all pages, equations, sections and bibliographical references refer to those in the revised paper.
>
> > *1. It would be better to formally define Signal Traversing Path and forward reference it from the introduction.*
>
> We have added the definition of the signal traversing path in **Proposition 2** and **forward referenced it in the second paragraph in the introduction.**
>
> > *2. It would be better if one states upfront that some external mechanism is used to extract the initial states of the coarser nodes. The clarity of section 3.2 can be improved.*
>
> We have briefly introduced what the nodes at each scale represent and further mentioned that the coarser-scale nodes are initialized using the coarser-scale construction module in the third paragraph in the introduction as follows:
>
> ''The inter-scale connections build a multiresolution representation of the original sequence: nodes at the finest scale correspond to the time points in the original time series (e.g., hourly observations), while nodes in the coarser scales represent features with lower resolutions (e.g., daily, weekly, and monthly patterns). **Such latent coarser-scale nodes are initially introduced via a coarser-scale construction module.**''
>
> To make **Section 3.2** more clear, we have added another sentence in this section:
>
> ''Specifically, the coarse-scale nodes are introduced scale by scale from bottom to top by performing convolutions on the corresponding children nodes $\mathbb C_\ell^{(s)}$.''
>
> We cannot add further explanations due to the page limit and hope the above one-sentence summary helps to clarify this section.
>
> > *3. Where is N defined? What is the relation between N and S ? I would suggest defining A, L, C, N, S clearly in one place with some indentation in Section 3.1, for ease of reading.*
>
> $N$ is defined under Eq. (3) in the paper. Note that $N$ is the number of attention layers and $S$ is the number of scales. We treat these two parameters as hyper-parameters. They can be chosen independently. We recommend to determine the number of attention layers $N$ based on the available computing resources, as this number is directly related to the model complexity. On the other hand, the number of scales $S$ can be determined by the granularity of the time series. For example, for hourly observations, we typically assume that it may also have daily, weekly and monthly periods. Therefore, we can set $S$ to be 4. We have added a discussion on hyper parameters selection in **Appendix K**.
>
> In addition, we notice that the number of notations used in this paper is relatively large. To avoid any further confusion, we have added a table in the Appendix that defines all notations (see **Table 4** in the paper).
>
> > *4. Please be slightly more clear in the second paragraph of section 3.3. I would like to verify my understanding of the multi-step forecasting modules. In the first module, we just have a fixed output layer which is a dense mapping to F outputs, where F is the future prediction length. In the second mapping, there is a decoder that sequentially generated the multi-step output and the decoder is equivalent to the decoder in the original transformer paper.*
>
> Thanks for pointing this out!
>
> In the first module, yes, we did apply a fixed fully-connected layer to map the outputs of the encoder to the $F$ time points to be predicted.
>
> The second prediction module is different from the one in the original Transformer though. In Transformer, the decoder outputs the predictions one by one in an autoregressive manner. By contrast, in Pyraformer, we output the multi-step prediction results in a batch.
>
> To solve your confusion, we have modified the second paragraph in **Section 3.3** as follows: (see the next part)

---

### Official Review · Reviewer_WMVh · 2021-11-01

**Correctness:** 4
**Technical Novelty And Significance:** 4
**Empirical Novelty And Significance:** 3
**Recommendation:** 6
**Confidence:** 4

**Main Review:**

Strengths
---
1.	The architecture is well-motivated, tackles the important problem of LSTF, and improves both forecasting performance and computational efficiency of state-of-the-art baselines.
2.	Paper is well written and easy to follow – making motivations and contributions clear.

Weaknesses
---
However, some key architectural details can be clarified further for full reproducibility and analysis. Specifically:
1.	How are historical observations combined with inputs known over all time given differences in sequence lengths (L vs L+M)? The text mentions separate embedding and addition with positional encoding, but clarifications on how the embeddings are combined and fed into the CSCM are needed.
2.	Can each node attend to its own lower-level representation? From equation 2, it seems to be that only neighbouring nodes are attended to, based on the description of N_l^(s).
3.	Do the authors have any guidelines on how to select S/A/C (and consequently N) for a given receptive field L?

In addition, while the ablation analysis tests the impact of changing CSCM architectures, it would be good to evaluate the base performance without the PAM to determine the value added by attention. This would also provide a simple comparison vs dilated CNNs which have been used successfully in time series forecasting applications  (e.g. WaveNet).

Finally, could I double check which dataset was used for the ablation analysis as well? I seem to be having some difficulty lining the numbers in Tables 4-6 up with Table 3.


**Summary Of The Paper:**

The authors propose a new architecture to tackle the problem of long sequence temporal forecasting (LSTF) – which looks at capturing long-range dependencies in time-series data by providing more direct paths between the output and distant history.

The Pyraformer takes an interesting spin on the current state-of-the-art sparse transformers, consisting of 2 main components:
1.	A dilated CNN encoder to learn coarse-scale representations at multiple resolutions.
2.	A decoder with a pyramid structure that applies attentions masks to a limited subset of nearest neighbours (i.e. parents, adjacent nodes and children) – effectively sparsifying fully connected attention patterns by imposing a prior structure onto attention patterns.


**Summary Of The Review:**

The paper makes a strong contribution to long sequence temporal forecasting, although there are some aspects that need to be verified before it is ready for publication.

---

> ### Author Response · Authors · 2021-11-22
> **Response to reviewer WMVh, part2**
>
> > *3. Do the authors have any guidelines on how to select S/A/C (and consequently N) for a given receptive field L?*
>
> We have added a discussion on hyper-parameter selection in **Appendix K**:
>
> ''We recommend to first determine the number of attention layers $N$ based on the available computing resources, as this number is directly related to the model size. Next, the number of scales $S$ can be determined by the granularity of the time series. For example, for hourly observations, we typically assume that it may also have daily, weekly and monthly periods. Therefore, we can set $S$ to be 4. We then focus on the selection of $A$ and $C$. According to the ablation study, we typically prefer a small $A$, such as 3 and 5. Lastly, in order to ensure the network has a receptive field of $L$, we can select a $C$ that satisfies Equation (5). In practice, we can use a validation set to choose $C$ from its candidates that satisfies (5). It is also worthwhile to check whether choosing different $C$ for different scales based on the granularity of the time series can further improve the performance as we did in Appendix I.''
>
> > *It would be good to evaluate the base performance without the PAM to determine the value added by attention.*
>
> Following your advice, we have added another ablation study in **Appendix J** to show the importance of the PAM and also checked the performance of Wavenet on the ETTm1 dataset. The experimental results are summarized as follows:
>
> | Method | Metrics | 96 | 288 | 672 |
> | ---------- | ---------- | --- | ----- | ----- |
> | CSCM Only | MSE | 0.576 | 0.782 | 0.883 |
> | CSCM Only | MAE | 0.544 | 0.683 | 0.752 |
> | Wavenet      | MSE | 0.571 | 0.766 | 0.937 |
> | Wavenet      | MAE | 0.526 | 0.649 | 0.747 |
> | Pyraformer  | MSE | **0.480** | **0.754** | **0.857** |
> | Pyraformer  | MAE | **0.486** | **0.659** | **0.707** |
>
> For a fair comparison, the number of parameters of the three methods were controlled to be within the same order of magnitude. In particular, we increased the bottleneck dimension of "Conv. w/bottleneck". We stacked 18 basic Wavenet blocks, and the resulting Wavenet has a receptive field of 190. Our experimental results show that the Pyraformer with the PAM yields much more accurate predictions than the remaining two methods, indicating the importance of the PAM. In addition, the CSCM itself performs slightly worse than Wavenet when the prediction length is 96 and 288, since the CSCM only stacks three convolution layers. However, when the prediction length is 672, the performance of CSCM exceeds that of Wavenet, since the receptive field of Wavenet is only 190 whereas that of the CSCM is 672.
>
> > *Finally, could I double check which dataset was used for the ablation analysis as well? I seem to be having some difficulty lining the numbers in Tables 4-6 up with Table 3.*
>
> We have provided more information regarding the experiment settings in the section of ablation study in **Apendix J**. **Note that the results are presented in Tables 7-9 in the updated version of this paper**.
>
> The ablation study on the impact of $A$ and $C$ (Table 7) was done on the ETTh1 dataset with the prediction length $M = 720$. On the other hand, in Table 3, $A = 5$ and $C = 4$ was our initial setting. However, in order to better show the impact of $A$, we did not include $A=5$ in Table 7. Therefore, the results in Table 3 cannot be found in Table 7.
>
> The ablation study on the impact of the CSCM architecture (Table 8) was also done on the ETTh1 dataset. To reduce the number of parameters, we used convolution with bottleneck in our long-range forecasting experiments. Thus, the performance of Pyraformer on ETTh1 with prediction length 168 in Table 3 is the same with the last row in Table 8.
>
> The ablation study on the impact of history length (Table 9) was done on the ETTm1 dataset, since its granularity is minute and contains more long-range dependencies. However, we are very sorry that we misstated the prediction length in Table 9. In order to better study the impact of history length, our prediction length is 1344. Therefore, both MSE and MAE in Table 9 are larger than that of Pyraformer on ETTm1 with the prediction length 672. We have corrected this mistake in the paper.
>
> To facilitate the reproduction of the results in the paper, we will also provide shell scripts corresponding to the experiments done in the paper when the code is made public.

---

> > ### Comment · Reviewer_WMVh · 2021-11-27
> > **Thank you for your response**
> >
> > Thank you for your detailed response, which has convincingly addressed my initial concerns. I think that this paper makes a strong contribution and will be updating my review to recommend acceptance.

---

> > > ### Author Response · Authors · 2021-11-30
> > > **Thanks for the Response of Reviewer WMVh**
> > >
> > > Many thanks to Reviewer WMVh again for providing a detailed valuable pre-rebuttal review. Your detailed suggestions help us a lot in paper revision.
> > >
> > > We'd also thank you for carefully judging our feedback and raising the score! Your constructive suggestions are very helpful for us to improve the paper in a better shape.

---

> ### Author Response · Authors · 2021-11-22
> **Response to reviewer WMVh, part1**
>
> Thanks very much for your detailed review and constructive feedback. We believe that addressing the reviewer's comments has resulted in improving the clarity and presentation of the paper’s contributions and has brought the paper to a higher standard. Below we address your points individually. Corresponding modifications in the paper are **highlighted in blue**. Unless otherwise stated in our response, all pages, equations, sections and bibliographical references refer to those in the revised paper.
>
> > *Some key architectural details can be clarified further for full reproducibility and analysis.*
>
> Thanks for your constructive suggestion. We have answered your questions about reproducibility below. We have also added more experimental details in the **Appendix F**.
>
> > *1. How are historical observations combined with inputs known over all time given differences in sequence lengths (L vs L+M)?*
>
> Recall that we have two prediction modules in the paper. Depending on the prediction module we use, the input to the CSCM is different.
>
> When using the first prediction module (i.e., the fully connected layer), we add an ''end token'' to the end of the historical observation sequence $z_{t-L+1:t}$. The value of the end token is 0 and the corresponding covariate is $x_{t+1}$. Thus, the embedding of $x_{t-L+1:t+1}$ is added with the observation embedding of $[z_{t-L+1:t}, 0]$ and positional embedding, and the resulting embedding with length $L+1$ is fed into the CSCM.
>
> When using the second prediction module (i.e., the attention-based decoder), the covariate sequence is first split into two parts: $x_{t-L+1:t}$ and $x_{t+1:t+M}$. The embedding of $x_{t-L+1:t}$ is added with the associated observation embedding and positional embedding, and the resulting embedding with length $L$ is fed into the CSCM. The remaining part of the covariate embedding $x_{t+1:t+M}$ is added with the corresponding positional embedding to construct the ''prediction tokens''. The prediction tokens are then fed into the decoder.
>
> We have added definitions of the end token and the prediction tokens in **Section 3.3**.
>
> > *2. Can each node attend to its own lower-level representation?*
>
> Yes, it can. In the definition of $A_l^{(s)}$ in Equation (2), we can see that each node can attend to the nodes that have a distance less than $(A-1)/2$ with it on the same scale. The distance between each node and itself is 0, so each node can attend to its own lower-level representation.
>
> We have explicitly explain this point in **the second paragraph on Page 5** as:
>
> ''each node in the graph can attend to a set of neighboring nodes $\mathbb N_\ell^{(s)}$ at three scales: the adjacent $A$ nodes at the same scale **including the node itself** (denoted as $\mathbb A_\ell^{(s)}$) ...''

---

### Official Review · Reviewer_f9f9 · 2021-11-02

**Correctness:** 3
**Technical Novelty And Significance:** 2
**Empirical Novelty And Significance:** 3
**Recommendation:** 6
**Confidence:** 4

**Main Review:**

Strength:
1. The paper is well written and the proposed architecture is easy to understand. I believe the equations are correct, and the included model illustrations makes the model design very intuitive.
2. Extensive evaluation is provided, and the proposed model consistently outperforms baseline models.
3. A ablation study is provided to justify the effectiveness of each component in the proposed architecture.

Weakness:
1. Although I couldn't find any specific publication, I believe hierarchical structure illustrated in Figure 1 (d) has been explored before in the context of LSTM/RNN. In addition, another line of work trying to apply the residual connections from ResNet to RNNs (e.g., https://arxiv.org/pdf/1701.03360.pdf) also provides a means for long-range time-series modelling.
2. The paper states that Pyraformer simultaneously capture temporal dependencies of different ranges in a compact multi resolution fashion. This is intuitively understandable. However, it would be more convincing if the evaluation could also illustrate this perspective. To be more specific, how do we know from the current evaluation setting that the model indeed learns multi resolution temporal dependencies?
3. Connected to comment 2, it would be better if the authors could illustrate a bit more on why these datasets can be used to evaluate long-term dependency modelling. In my understanding, long-term dependency modelling could be a bit different from long-term prediction. The former focuses on extract meaningful information from long-term previous steps, and the latter focuses on accurate prediction into the far future. If the real-world dataset does not include the long-term dependency in it, some study on a synthetic dataset would help readers understand the benefit of the proposed model. In other words, inaccuracy in long-term prediction could be resulted from (1) accumulated noise in predictions along time, and (2) lack of long-term temporal dependency modelling. How does the authors distinguish the result from these two perspectives?
4. The proposed method uses a factor of 2 to construct the hierarchy. This number seems a bit arbitrary. I'm wondering if this is a design choice with some thoughts behind it, or could this number be treated as a hyper parameter? Basically this number will decide the number of hierarchies in the structure.

**Summary Of The Paper:**

This paper proposes Pyraformer, a low-complexity pyramidal attention model for long-range time-series modelling and forecasting. The proposed architecture is build upon a pyramidal attention module (PAM) in which the inter-scale tree structure summarizes features at different resolutions and the intra-scale neighbouring connections model the temporal dependencies of different ranges. The proposed framework harnesses the benefit of both transformer and RNN. Evaluation on Electricity, wind, App Flow, and ETT dataset against baseline models including Informer., LogTrans, Longformer, Reformer, ETC shows superior performance.

**Summary Of The Review:**

Overall it is a solid paper. Please refer to the above session for the detailed discussion. I had a few concerns regarding evaluation setup and novelty. If the authors could reply accordingly, I'm willing to reevaluate the paper.

---

> ### Author Response · Authors · 2021-11-22
> **Response to reviewer f9f9, part2**
>
> > *3. It would be better if the authors could illustrate a bit more on why these datasets can be used to evaluate long-term dependency modeling. If the real-world dataset does not include the long-term dependency in it, some studies on a synthetic dataset would help readers understand the benefit of the proposed model.*
>
> Thanks a lot for your constructive suggestion!
>
> The three datasets used for long-range forecasting in this paper, i.e., ETTh1, ETTm1 and Electricity, are also used in Informer (Zhou et.al, 2021), a recent work on long-range forecasting, to show the superiority of Informer in long-range forecasting tasks. Hence, we believe it is appropriate to use these datasets to evaluate the proposed Pyraformer.
>
> On the other hand, we agree that long-term dependency modeling is a bit different from long-term prediction. However, we would like to argue that capturing long-term dependency helps to yield accurate long-term forecasting results.
>
> In addition, all benchmark methods implemented in this paper obtain multi-step long-range prediction results together in a single forward operation instead of in an autoregressive step-by-step manner. Therefore, there is no error accumulation over time. We can safely assume that the prediction error mainly arises from the lack of long-term temporal dependency modeling.
>
> Since it is difficult to tell whether long-term dependency exists in real-world datasets, we follow your suggestion and generate a synthetic dataset with long-range dependencies. Detailed synthesis formula and experimental results are shown in **Appendix I**. Here we briefly illustrate how we generate the synthetic dataset and show the main experimental results.
>
> Specifically, each time series in the synthetic dataset is a linear combination of three sine functions of different periods: 24, 168 and 720, that is,
>
> \begin{equation}
>     f(t)= \beta_0 + \beta_1 \sin(\frac{2\pi}{24}t) + \beta_2 \sin(\frac{2\pi}{168}t) + \beta_3 \sin(\frac{2\pi}{720}t).
> \end{equation}
>
> In the above equation, the coefficients of the three sine functions $\beta_1$, $\beta_2$, and $\beta_3$ for each time series are uniformly sampled from $[5, 10]$. $\beta_0$ is a Gaussian process with a covariance function $\Sigma_{t_1, t_2} = |t_1 - t_2|^{-1}$ and $\Sigma_{t_1} = \Sigma_{t_2} = 1$, where $t_1$ and $t_2$ denote two arbitrary time stamps. Such polynomially decaying covariance functions are known to have long-range dependence. In our experiments, we use the historical 720 time points to predict the future 720 points. **Since both the deterministic and stochastic parts of the synthetic time series have long-range correlations, such dependencies should be well captured in the model in order to yield accurate predictions of the next 720 points.** The results are summarized below.
>
> | Method         |    MSE   |   MAE   |
> | ----------------- | ---------- | ---------- |
> | Full attention|    3.550  |   1.477  |
> | LogTrans     |   3.007   |   1.366  |
> | ETC             |   4.742   |   5.509  |
> | Informer       |    7.546  |    2.092 |
> | Longformer  |    2.032 | 1.116 |
> | Reformer     | 1.538 | 3.069 |
> | Pyraformer   | 1.258 | 0.877 |
> | Pyraformer-period | 1.176 | 0.849 |
>
> It can be observed that Pyraformer outperforms the benchmark methods by a large margin. In particular, the MSE given by Pyraformer is decreased by **18.2%** compared with Reformer, which produces the smallest MSE among the existing variants of Transformer. The performance of Pyraformer-period will be discussed in the next reply.
>
> > *4. The factor to construct the hierarchy.*
>
> We did not use a binary tree in our experiments. The binary tree in Figure 1 is schematic only. In **Section 3.1**, we denote the number of children for each node in the coarser scales as $C$ and further provide some guidelines to choose $C$ in Proposition 2 in order to obtain a short signal traversing path. In practice, we can choose $C$ using two strategies. First, we can regard $C$ as a hyper parameter, and determine its value from the candidates that satisfy Equation (5) using a validation set. Second, we can choose $C$ based on the granularity of the observations. For instance, for hourly observations, we typically assume that it may also have daily, weekly and monthly periods. As a result, we can set $C = 12, 7, 4$ sequentially for each scale from bottom to top, and so the nodes in coarser scales represent half a day, half a week, and half a month, respectively. The result of this configuration is referred to as ''**Pyraformer-period**'' in the above table. In comparison with the Pyraformer with the same $C$ for all scales (i.e., Pyraformer in the above table), the performance of Pyraformer-period can be further improved. We have included this discussion in **Appendix I**. Besides, we also added a discussion about the selection of hyper-parameters in **Appendix K**.

---

> ### Author Response · Authors · 2021-11-22
> **Response to reviewer f9f9, part1**
>
> We value your constructive and thoughtful comments. We believe that addressing the reviewer's comments has resulted in improving the clarity and presentation of the paper’s contributions and has brought the paper to a higher standard. Below we address your concerns point by point. Corresponding modifications in the paper are **highlighted in blue**. Unless otherwise stated in our response, all pages, equations, sections and bibliographical references refer to those in the revised paper.
>
> > *1. Hierarchical structure illustrated in Figure 1 (d) has been explored before in the context of LSTM/RNN.*
>
> Thanks for pointing this out! We have added a brief review on related RNNs and LSTMs in the **Appendix A** and discussed their differences from Pyraformer as follows. **We further refer the readers to Appendix A in the second paragraph in Section 2.1**.
>
> ''In this section, we provide a brief review on the related RNN-based models. Multiscale temporal dependencies are successfully captured in HRNN (Costa-juss`a \& Fonollosa, 2016) and HM-RNN (Chung et al., 2019).  The former requires expert knowledge to partition the sequence into different resolutions, while the latter learns the partition automatically from the data. Note that the theoretical maximum length of the signal traversing path in both models is still $\mathcal O(L)$.   Another line of works aim to shorten the signal traversing path by adding residual connections (Kim et al., 2017) or dilated connections to LSTMs (Chang et al., 2017). However, they do not consider the multiresolution temporal dependencies explicitly. Furthermore, all aforementioned RNNs only propagate information in one direction from the past to the future.  An appealing approach that allows bidirectional information exchange is Bi-LSTM (Schuster, 1996). The forward and backward propagation is realized through two different LSTMs though, and so still incurs a long signal traversing path.
>
> As opposed to the above mentioned LSTM/RNN models, the proposed Pyraformer enables bidirectional information exchange that can better model the temporal dependencies, while providing a multiresolution representation of the observed sequence at the same time. We also notice that due to the unidirectional property of RNNs, it is difficult the realize the pyramidal graph in Figure 1d based on RNNs.''
>
> > *2. It would be more convincing if the evaluation could also illustrate that Pyraformer simultaneously capture temporal dependencies of different ranges in a compact multi resolution fashion.*
>
> We completely agree that it is better if we can show that Pyraformer indeed learns multiresolution temporal dependencies. To move forward to this goal, we have depicted the extracted features for an arbitrary chosen channel in **Figure 7**. It can be observed from the figure that features at the coarser scales can be regarded as a lower resolution version of the features at the finer scales.

---

> ### Comment · Reviewer_f9f9 · 2021-11-28
> **Raising my recommendation score base on rebuttal**
>
> I want to thank the authors for providing a rebuttal discussing my concerns on the paper. Overall I believe the rebuttal is thorough and resolved most of my concerns. Therefore, I'm raising my recommendation score.

---

> > ### Author Response · Authors · 2021-11-30
> > **Thanks for the Response of Reviewer f9f9**
> >
> > We'd like to thank Reviewer f9f9 again for providing an insightful pre-rebuttal review, which has enabled us to make an effective response. Your detailed suggestions help us a lot.
> >
> > We'd also thank you for carefully judging our feedback and raising the score! Your constructive suggestions are very helpful for us to improve the paper in a better shape.

---

### Official Review · Reviewer_cdAe · 2021-11-03

**Correctness:** 4
**Technical Novelty And Significance:** 3
**Empirical Novelty And Significance:** 3
**Recommendation:** 8
**Confidence:** 4

**Main Review:**

## Strengths
1. The paper is well-written and well-motivated with sufficient technical details
2. The extensive empirical results demonstrate the effectiveness and efficiency of the proposed method
3. A proof is provided to guarantee the linear complexity of long sequence encoding

## Comments
1. More datasets for multi-step forecasting would be helpful in evaluating the method.
2. The reason of using the second prediction module in multi-step forecasting need more justification, e.g. why take the concatenation as key/value in the second layer.

**Summary Of The Paper:**

This paper presents a new hierarchical transformer architecture with constant connection path length and linear time and space complexity for long-range time series modeling. The module at core is a pyramidal attention network that makes multi-resolution representations in a tree structure and perform attention operations on the tree. A stack of convolutions is used to initialize the pyramidal tree. Experiments show the proposed method is able to make more accurate predictions with significantly fewer attention operations and, as a result, less time and memory expenses.

**Summary Of The Review:**

Overall I find this paper quite interesting with great potential contribution to the community, and recommend an accept.

---

> ### Author Response · Authors · 2021-11-22
> **Response to reviewer cdAe**
>
> Many thanks for your positive review and valuable feedback!  We believe that addressing the reviewer’s comments has resulted in improving the clarity and presentation of the paper’s contributions and has brought the paper to a higher standard. Below we address your comments individually. The corresponding modifications in the paper are **highlighted in blue**. Unless otherwise stated in our response, all pages, equations, sections, and bibliographical references refer to those in the revised paper.
>
> > *1. More datasets for multi-step forecasting would be helpful in evaluating the method.*
>
> Thank you for your suggestion. We further synthesized a dataset with long-range dependence and carried out experiments on it to verify the long-range modeling capability of the proposed method. We presented the details about the dataset and experimental results in **Appendix I**. Here we briefly illustrate how we generate the synthetic dataset and show the main experimental results.
>
> Specifically, each time series in the synthetic dataset is a linear combination of three sine functions of different periods: 24, 168 and 720, that is,
>
> \begin{equation}
>     f(t)= \beta_0 + \beta_1 \sin(\frac{2\pi}{24}t) + \beta_2 \sin(\frac{2\pi}{168}t) + \beta_3 \sin(\frac{2\pi}{720}t).
> \end{equation}
>
> In the above equation, the coefficients of the three sine functions $\beta_1$, $\beta_2$, and $\beta_3$ for each time series are uniformly sampled from $[5, 10]$. $\beta_0$ is a Gaussian process with a covariance function $\Sigma_{t_1, t_2} = |t_1 - t_2|^{-1}$ and $\Sigma_{t_1} = \Sigma_{t_2} = 1$, where $t_1$ and $t_2$ denote two arbitrary time stamps. Such polynomially decaying covariance functions are known to have long-range dependence. In our experiments, we use the historical 720 time points to predict the future 720 points. Since both the deterministic and stochastic parts of the synthetic time series have long-range correlations, such dependencies should be well captured in the model in order to yield accurate predictions of the next 720 points. The results are summarized below.
>
> | Method         |    MSE   |   MAE   |
> | ----------------- | ---------- | ---------- |
> | Full attention|    3.550  |   1.477  |
> | LogTrans     |   3.007   |   1.366  |
> | ETC             |   4.742   |   5.509  |
> | Informer       |    7.546  |    2.092 |
> | Longformer  |    2.032 | 1.116 |
> | Reformer     | 1.538 | 3.069 |
> | Pyraformer   | 1.258 | 0.877 |
>
> It can be observed that Pyraformer outperforms the benchmark methods by a large margin. In particular, the MSE given by Pyraformer is decreased by **18.2%** compared with Reformer, which produces the smallest MSE among the existing variants of Transformer. For more details about the synthetic dataset and visualization results, please refer to **Appendix I**.
>
> > *2. The reason of using the second prediction module.*
>
> We concatenate $\mathbf F_e$ and $\mathbf F_{d1}$ in the second layer instead of only using $\mathbf F_{d1}$ since we believe that the historical information $\mathbf F_e$ is essential for accurate long-range forecasting.
>
> We have added more justifications for the second prediction module in **Section 3.3** as follows:
>
> ''The second one, on the other hand, resorts to a decoder with two full attention layers. Specifically, **similar to the original Transformer (Vaswani et al., 2017)**, we replace the observations at the future $M$ time steps with 0, embed them in the same manner with the historical observations, and refer to the summation of the observation, covariate, and positional embedding as the ''prediction token'' $\mathbf{F_p}$. The first attention layer then takes the prediction tokens $\mathbf{F_p}$ as the query and the output of the encoder $\mathbf{F_e}$ (i.e., all nodes in the PAM) as the key and the value, and yields $\mathbf F_{d1}$. The second layer takes $\mathbf{F_{d1}}$ as the query, but takes the concatenated $\mathbf{F_{d1}}$ and $\mathbf F_e$ as the key and the value. **The historical information $\mathbf F_e$ is fed directly into both attention layers, since such information is vital for accurate long-range forecasting.** The final prediction is then obtained through a fully connected layer across the dimension of channels. **Again, we output all future predictions together to avoid the problem of error accumulation in the autoregressive decoder of Transformer.**''

---

### Public Comment · ~Zihao_Ye1 · 2022-01-30
**A suggestion on related work.**

Dear authors,

Congratulations on the acceptance, I was the author of BP-Transformer and I noticed you mentioned "Note that BP-Transformer initializes the nodes at each scale by embedding the corresponding sequences" in the related work section. This argument is not true because BP-Transformer initializes span nodes with all zeros and leaf nodes with corresponding word embedding. I hope you can consider revising this argument.

Thanks

---

> ### Public Comment · ~Hang_Yu1 · 2022-01-30
> **Noted with thanks!**
>
> Thanks for pointing this out! We will correct the argument in the camera-ready version.

---

### Public Comment · ~Yi_Rao2 · 2022-09-02
**The comparison with CNN is not fair**

The maximum path length of CNN is $ O(log_kL) $ instead of $ O(L) $ as written in Table 1, where k is the kernel width. if $ k ∝ \sqrt[S-1]{L} $ is chosen (the same way as C is chosen in the paper), then the maximum path length of CNN should also be $ O(1) $. The complexity per layer & maximum path length of Pyraformer is actually the same as that of CNN, and Pyraformer is not proven to be superior to CNN in the paper.

---

> ### Public Comment · ~Shizhan_Liu1 · 2022-09-05
> **Response to Yi Rao**
>
> Thank you for your valuable comment. What we refer to in Table 1 is the general convolution. As mentioned in [1], the maximum path length for the general convolution is $O(\frac{L}{k})$, while that for dilated convolution is $O(log_k⁡ L)$:
>
>     Doing so requires a stack of O(n/k) convolutional layers in the case of contiguous kernels,
>     or O(log_k(n)) in the case of dilated convolutions, increasing the length of the longest paths
>     between any two positions in the network.
>
> For the dilated convolution, we have also compared the performance of Pyraformer with Wavenet in the experiment, see 'Response to reviewer WMVh, part2'. Pyraformer performs a little bit better than Wavenet. This may be because Pyraformer passes information through the pyramidal graph in each layer, while the dilated convolution loses some local information in order to expand the receptive field.
>
> In addition, the CSCM of Pyraformer can not only use convolution, but also use parameterless operators such as average pooling, Max pooling, etc. This may help to alleviate the problem of large parameters and hard to optimize when the convolution kernel is large.
>
> [1] Ashish Vaswani, Noam Shazeer, Niki Parmar, Jakob Uszkoreit, Llion Jones, Aidan N Gomez,
> Łukasz Kaiser, and Illia Polosukhin. Attention is all you need. In Advances in neural information
> processing systems, pp. 5998–6008, 2017.

---

> > ### Public Comment · ~Yi_Rao2 · 2022-09-06
> > **Response to Shizhan Liu**
> >
> > Thank you for your reply, so it's clear that the complexity per layer & maximum path length of Pyraformer is the same as that of dilated CNN.
> >
> > And I noticed that in your code you shuffle the training data, I know that this is the setting used by Informer and its followers, but it doesn't mean that this is the correct setting. In fact, in practice, we should not shuffle the data when training for a time series prediction problem, because it can lead to serious leakage (using future data to predict the past). Not to mention that shuffle makes the PositionalEmbedding  in your code meaningless during training, because the order you use as position is actually disordered. If you compared the performance of Pyraformer with Wavenet or other models under this setting, the comparison may not be correct, as some models may overfit the training set and resulting in worse test results due to this leakage.

---

> > > ### Public Comment · ~Shizhan_Liu1 · 2022-09-06
> > > **Response to Yi Rao (2)**
> > >
> > > I don't think shuffling the training data will cause future information leakage.
> > >
> > > First, operations in the network are parallel in the 'batch' dimension and do not model the correlations between the different time series within the batch. Thus, even if the future time series is in the same batch as the past time series, the future information is not leaked to the past.
> > >
> > > Second, assuming that the network trained on the training set predicts the future based on future information leakage, then it should do poorly on the test set, because there is no shuffle in the test set. However, many methods perform better on the training set with shuffle than without shuffle.
> > >
> > > Third, if information leakage occurs when the future time series and the past time series are in the same batch, then training without shuffle will cause more leakage than that with shuffle. Since the time series in the same batch are closer in time and have higher correlation when no shuffle is done.
> > >
> > > Finally, shuffling the training dataset did not start with Informer. For example, as early as in 2017, DeepAR [1] was not trained on the unshuffled training set.
> > >
> > > [1] Salinas, D., Flunkert, V., & Gasthaus, J. (2017). DeepAR: Probabilistic Forecasting with Autoregressive Recurrent Networks. arXiv preprint arXiv:1704.04110.

---

### Public Comment · ~Nikolaos_Kourentzes1 · 2022-09-26
**Potentially helpful work from the time series forecasting literature**

Interesting work and glad to see its good performance!

There is a similar idea that has been developing in the time series forecasting literature for the past years. It may help you in connecting some of the intuitions there with your work. Here is the paper that introduced the modelling idea:
- Kourentzes, N., Petropoulos, F., & Trapero, J. R. (2014). Improving forecasting by estimating time series structural components across multiple frequencies. International Journal of Forecasting, 30(2), 291-302.

And a follow up work that made the theory quite a bit more solid:
- Athanasopoulos, G., Hyndman, R. J., Kourentzes, N., & Petropoulos, F. (2017). Forecasting with temporal hierarchies. European Journal of Operational Research, 262(1), 60-74.

There are quite a few papers building on these, but I think these are a good start, if you find them useful. It is always difficult to keep track of what is happening in adjacent fields!

---

> ### Public Comment · ~Shizhan_Liu1 · 2022-09-27
> **Noted with thanks!**
>
> Thank you for your attention to our work and valuable suggestions! We'll take a closer look at the literature you have listed.

---

### Public Comment · ~贻钦_张1 · 2023-05-10
**There may be a tiny-mistake in your formula**

In formula 3, the subscript of the summation symbol in the denominator seems to be problematic.

the original is:
$\sum\limits_{ ℓ \in {\scriptstyle \mathbb{N}}_{\scriptstyle l}^{\scriptstyle (s)}}$

i believe is:
$\sum\limits_{ ℓ \in {\scriptstyle \mathbb{N}}_{\scriptstyle ℓ}^{\scriptstyle (s)}}$

If there are something that i mis-understand, please point out, tks~

---

> ### Public Comment · ~Shizhan_Liu1 · 2023-05-11
> **Noted with thanks!**
>
> Thanks for pointing this out. The subscript of $\mathbb{N}$ should be $\ell$, as you wrote.

---

### Decision · Program_Chairs · 2022-01-20

**Decision:**

Accept (Oral)

**Comment:**

The authors propose a multi-resolution pyramidal attention mechanism to capture long-range dependencies in time series forecasting, achieving linear time and space complexity. The authors conduced an extensive set of experiments and ablation studies demonstrating that  the proposed method consistently outperforms the state-of-the-art and provided evidence for the various components of the architecture. They also provided a proof guarantee the linear complexity of long sequence encoding and adequately addressed the concerns raised by the reviewers. The additional additional benchmarks conducted by the author further demonstrated the strong performance of the method. All reviewers agreed that this work makes a solid contribution to the field.